# Sex differences in functional cortical organization reflect differences in network topology rather than cortical morphometry

Bianca Serio [1,2,3,4] ✉, Meike D. Hettwer [1,2,3,4], Lisa Wiersch [1,2,5], Giacomo Bignardi[3,6], Julia Sacher[3,4,7,8], Susanne Weis [1,2], Simon B. Eickhoff[1,2,3] & Sofie L. Valk [1,2,3,4] ✉

Differences in brain size between the sexes are consistently reported. However, the consequences of this anatomical difference on sex differences in intrinsic brain function remain unclear. In the current study, we investigate whether sex differences in intrinsic cortical functional organization may be associated with differences in cortical morphometry, namely different measures of brain size, microstructure, and the geodesic distance of connectivity profiles. For this, we compute a low dimensional representation of functional cortical organization, the sensory-association axis, and identify widespread sex differences. Contrary to our expectations, sex differences in functional organization do not appear to be systematically associated with differences in total surface area, microstructural organization, or geodesic distance, despite these morphometric properties being per se associated with functional organization and differing between sexes. Instead, functional sex differences in the sensory-association axis are associated with differences in functional connectivity profiles and network topology. Collectively, our findings suggest that sex differences in functional cortical organization extend beyond sex differences in cortical morphometry.

Sex differences in human global brain size are robust and widely acknowledged[1–7], but the functional implications of this anatomical difference are not well understood. Indeed, sex differences in intrinsic brain function are sometimes deemed small or negligible beyond differences attributed to brain size[2]. Nevertheless, diverging patterns of functional connectivity between males and females have been reported even when controlling for differences in brain size and most consistently in sensory and association regions[5,8,9]. These regions in fact represent the two anchors of a key principle of hierarchical functional organization, the sensory-association (S-A) axis, differentiating localized primary sensory/motor areas from a more distributed set of transmodal association regions, including regions belonging to the frontoparietal and default mode networks (DMN)[10,11]. However, the extent to which sex differences in intrinsic functional cortical organization may be explained by neuroanatomical differences relating to brain size remains unclear.

In order to understand how sex differences in brain size may pertain to sex differences in brain function, it is first necessary to understand the relevance of brain size for overall functional cortical organization. Brain size and its variability may have important

---

[1]Institute of Neuroscience and Medicine, Brain & Behavior (INM-7), Research Centre Jülich, Jülich, Germany. [2]Institute of Systems Neuroscience, Medical Faculty, Heinrich-Heine-Universität Düsseldorf, Düsseldorf, Germany. [3]Max Planck School of Cognition, Leipzig, Germany. [4]Max Planck Institute for Human Cognitive and Brain Sciences, Leipzig, Germany. [5]Brain-Based Predictive Modeling Lab, Feinstein Institutes for Medical Research, Glen Oaks, New York, NY, USA. [6]Language and Genetics Department, Max Planck Institute for Psycholinguistics, Nijmegen, The Netherlands. [7]Leipzig Center for Female Health & Gender Medicine, Medical Faculty, University Clinic Leipzig, Leipzig, Germany. [8]Clinic for Cognitive Neurology, University Medical Center Leipzig, Leipzig, Germany. ✉e-mail: b.serio@fz-juelich.de; valk@cbs.mpg.de

consequences for the spatial distribution of sensory and association areas across the cortical mantle, as illustrated by clear scaling patterns over evolution[12,13] and development[14]. In fact, over the past 4 million years, hominin evolution has not only shown a general trend of increasing body mass, but also an even more important relative increase in brain size[12]. According to the tethering hypothesis, the brain's sensory systems, acting as anchors, may have exerted constraining pressures during the growth of the developing ancestral mammalian cortex[13]. In this way, evolutionary cortical expansion may have led to the emergence of the S-A axis, with association cortices distributed across the cortical mantle and untethered from sensory hierarchies. Patterns of expansion across cortical regions along the S-A axis are also observed across human development, with a more markedly distributed areal expansion across frontoparietal association regions relative to limbic and sensorimotor areas[14]—see also[15] for a comprehensive review of the S-A axis' neurodevelopment. Through the increase of overall brain size, the differential expansion of sensory and association areas could thus be an important product of mammalian evolution and development, further reflecting the hierarchical functional differentiation of these regions. In fact, this different scaling and reorganization of regions along the S-A axis also appears to reflect patterns of sex differences in cortical morphometry, that is, cortical shape and size.

Morphometric differences between male and female brains have been extensively reported, with males showing a greater absolute brain volume[6]. Different measures of brain size are commonly used in the literature (such as intracranial volume, total brain volume, and total surface area). Although these measures highly covary and are often used interchangeably, they quantify different morphometric features of the brain, with sex differences in "brain size" ranging from 8% to 13% depending on the selected measure[6]. The size and direction of sex effects also vary by neuroanatomical property, such as different tissue types, brain regions, and features (including cortical thickness, gyrification, and surface area)[16]. It must be noted that within-group variability in cortical morphometry –which is typically greater in males– is larger than between-group mean effects, meaning that individual differences within sex are larger than group-differences between sexes[17]. Although individual differences in total brain size seem to account for most differences in relative regional volumes[3], some sex differences still remain statistically significant when the variance explained by total brain size is taken into account[7]. Therefore, there may be sex differences in the scaling of regional brain volume that go beyond linear associations with overall brain and body size. In fact, although local sex differences in cortical morphometry are typically small in size and vary across studies[18], they have been reported in both sensory and association regions: A meta-analysis identified volumetric sex differences in multiple cortical regions, including the anterior and posterior cingulate gyri, precuneus, right frontal pole, inferior and middle frontal gyri, insular cortex, Heschl's gyrus, and lateral occipital cortex[6]. Another study found greater gray matter volume in females in prefrontal and superior parietal cortices, whilst males showed greater volumes in ventral occipitotemporal regions[19]. These findings further depict the apparent relevance of the S-A axis as an axis of morphometric variability between the sexes. Developmental trajectories of anatomical change also appear regionally heterogeneous along the S-A axis, with higher rates of global cortical thickness change found in fronto-temporal association regions and lower rates found in sensory regions[20]. Morphometric cortical properties therefore seem to not only follow patterns of variation along the S-A axis, but also differ between the sexes. Yet, how exactly sex-specific differences in cortical morphometry may be relevant to differences in intrinsic brain function has not been directly explored.

Consistent with patterns of morphometric variation and sex differences, evidence points to sex differences in intrinsic functional connectivity (FC) in regions represented at the poles of the S-A axis[5,8,9].

In fact, despite generally controversial findings on sex differences in brain function, findings of stronger FC in females within the DMN[5,21–24] and stronger FC in males within sensorimotor areas[5,22,25] are consistent. Overlapping morphometric and functional patterns of sex differences along the S-A axis thus suggest that differentiation in functional cortical organization may be somewhat orchestrated by the cortical mantle's morphometric properties. Indeed, the structure, size, and shape of the cortex not only physically support functional connections, but also determine their length. Short- and long-range connections, as measured by geodesic distance (the distance separating two regions along the cortical mantle) have in fact been found in sensory and association regions respectively[26], thus also displaying patterns of variation along the S-A axis. With increasing distance between regions, cortical function also appears to change more rapidly in association regions relative to sensorimotor regions[27]. These patterns further mirror patterns of microstructural cortical variability identified by post-mortem histology[26] and myelin-sensitive in vivo magnetic resonance imaging (MRI)[26,27]. As such, intrinsic functional activity, showing variability between the sexes and along the S-A axis, seems to be embedded within the cortical mantle and its microstructural organization. Accumulating evidence further supports the important role played by cortical geometric properties, including size and shape, in sculpting functional architecture. Established findings from graph theory suggest that a cortical functional network's properties are largely determined by its spatial embedding, namely by the length of its connections[28]. Peaks of DMN clusters on the S-A axis also appear to be equidistantly distributed relative to primary areas[10], in line with the hypothesized untethering of association cortices from sensory hierarchies during evolutionary expansion[13]. Furthermore, recent findings suggest that the spatial organization of intrinsic cortical functional activity is dominated by long wave-lengths of geometric eigenmodes[29]. This research builds on notions from neural field theory positing that brain shape physically constrains brain-wide functional dynamics by imposing boundaries on emerging functional signals[30,31]. In the context of sex differences in functional cortical organization, brain size also explains some—although not all—sex-specific variance in FC[32]. Together, these findings point to possible morphometric properties that may not only underpin cortical functional architecture, but also be at the root of sex differences in functional cortical organization.

In the current work, we therefore investigated the extent to which sex differences in intrinsic functional cortical organization may be associated with differences in cortical morphometry, namely different measures of brain size, microstructure, and the geodesic distance of connectivity profiles. To this end, we used multimodal imaging data (including resting state functional MRI and structural T1 and T2 images) of the Human Connectome Project (HCP) S1200 release[33], consisting of healthy young adults self-reporting their biological sex. We began by computing the S-A axis as our measure of functional organization, given that it reflects multimodal mechanisms bridging morphometric, structural, and functional features of cortical organization, and given that sex differences in functional connectivity are typically found in regions situated at the extremities of this axis, i.e., in sensory and association regions. We then tested for sex differences along this low dimensional axis of hierarchical organization. Next, we identified the cortical morphometric properties potentially constraining the S-A axis, including different measures of brain size (recognizing total surface area as the most pertinent to the S-A axis), microstructural organization (a low dimensional microstructure profile covariance (MPC) axis), and the mean geodesic distance of connectivity profiles. We then probed associations between patterns of sex differences in cortical morphometry and patterns of sex differences in the S-A axis. Contrary to our expectations, we do not find evidence supporting a morphometric explanation of sex differences in functional organization, despite identifying sex differences in the morphometric properties per se. As such, we further probed potential functional features that may intrinsically

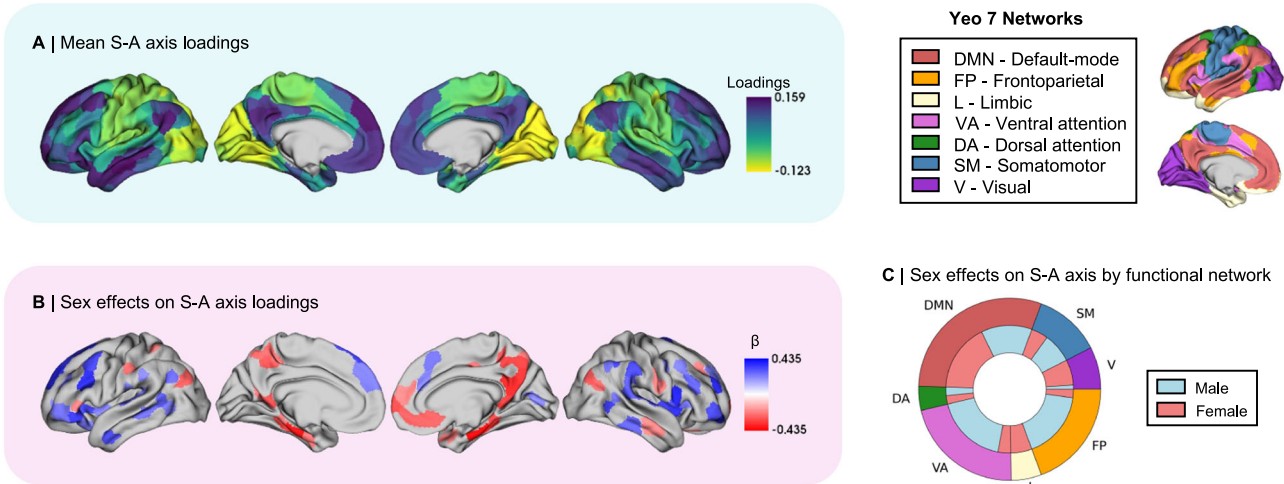

**Fig. 1 | The sensory-association (S-A) axis of functional cortical organization and its sex differences. A** Mean S-A axis loadings (spanning from visual to default-mode network regions) across sexes; **B** Thresholded β-map of linear mixed effect model (LMM) results showing false discovery rate (FDR)-corrected ($q < 0.05$) statistically significant effects of sex on S-A axis loadings, where blue represents higher male loadings and red represents higher female loadings; **C** Functional network breakdown of cortical areas showing statistically significant FDR-corrected sex differences in S-A axis loadings. The outer ring displays absolute proportions of statistically significant cortical areas by functional Yeo network, the inner ring displays absolute nested proportions by directionality of effects, where blue represents higher male loadings and red represents higher female loadings. β standardized beta coefficient. Source data are provided as a Source Data file.

underpin sex differences on the S-A axis, and our findings suggest that differences in FC profiles and network topology may be a more plausible explanation of sex differences in functional organization.

## Results

### Sex differences in the S-A axis of functional cortical organization

In order to investigate whether sex differences in functional organization may be related to sex differences in cortical morphometry, we began by constructing our measure of functional organization and testing for related sex differences. We thus computed the S-A axis at the individual level as our measure of functional organization in subjects of the HCP S1200 release[33]. For this, we applied a non-linear dimensionality reduction algorithm on FC Fisher r-to-z transformed matrices. We only considered the top 10% of the row-wise $z$ values, representing each seed region's top 10% maximally functionally connected regions[34,35]. We used this 90% threshold for consistency with previous studies[10,36,37] and for its high test-retest reliability and reproducibility[38,39]. As such, we computed the well-replicated low dimensional axis of functional brain organization explaining the most variance in the data (21.86%)—spanning from unimodal (sensory, here particularly visual) regions to heteromodal (association) regions[10]—and defined it as the S-A axis (Fig. 1A). Mean cortex-wide spatial patterns of S-A axis loadings were overall highly correlated between the sexes, $r_{spin} = 0.996$, $p_{spin} < 0.001$. Then, to test for regional effects of sex on S-A axis loadings, we fitted a linear mixed effects model (LMM) including fixed effects of sex, age, and total surface area, and random nested effects of family relatedness and sibling status (see Methods for more information on the nested structure of the HCP data and the statistical modeling). We identified sex differences in the S-A axis loadings in 23.3% of cortical regions (93 out of 400 Schaefer parcels) with small to medium effect sizes (minimum effect size of β = 0.210; maximum effect size of β = 0.435), which were distributed across the seven intrinsic functional Yeo networks[40] (Fig. 1B, C). Positive standardized beta coefficients (β values), depicted in blue, represent higher loadings in males relative to females on the S-A axis, whereas negative β-values, depicted in red, represent higher loadings in females relative to males. In Supplementary Fig. 1, we show that patterns of within-sex variability in S-A axis loadings are similar between males and females, with only a few regions showing statistically significant sex differences in variance.

### Morphometric correlates of the S-A axis

We then investigated potential morphometric constrains on functional organization by probing associations between the S-A axis and different measures of brain size, microstructural organization, and the mean geodesic distance of connectivity profiles.

First, we tested for associations between the S-A axis loadings and three measures of brain size commonly used in the literature, namely intracranial volume (ICV), total cortical volume (TCV), and total surface area. More specifically, ICV represents the entire volume encapsulated by the cranium (i.e., including cerebrospinal fluid), TCV represents the total volume of gray and white matter within the neocortex (excluding subcortical structures), and total surface area represents the entire surface area of the neocortical mantle (see Methods for the exact computation of these measures). Sex differences in these measures of brain size and other anthropometric measurements (i.e., height, weight, and body mass index) are further reported in Supplementary Table 1. For each measure of brain size, we fitted an LMM to test for regional effects of brain size on S-A axis loadings (Supplementary Fig. 2), and we found total surface area to have the most widespread effects amongst the three tested brain size measures (Fig. 2D; Supplementary Fig. 2C).

Second, we computed the microstructural profile covariance (MPC) axis of organization at the individual level, which is a low dimensional representation of the similarity of microstructural intensity profiles, defined as the T1-weighted (T1w) over T2-weighted (T2w) tissue intensity ratio, across cortical regions and layers[37,41,42]. We computed the MPC axis by conducting nonlinear dimensionality reduction on MPC matrices[34,35], which were obtained by sampling and correlating the intracortical microstructural intensity of 12 equivolumetric depth profiles (see Methods). Following the same approach used for computing the S-A axis, we selected the axis explaining the most variance in the data (25.81%)—spanning from sensory to paralimbic regions— defining it as the MPC axis (Fig. 2B). We specifically selected this low-dimensional representation of microstructural organization as it has been previously shown to covary with the low-dimensional representation of functional organization (i.e., the S-A axis)[42]. To test for whole-brain associations between the S-A and MPC axes, we correlated the spatial maps of the axes' mean loadings (Fig. 2A, B) across all subjects (Fig. 2G; $r = 0.20$, $p_{spin} = 0.037$). We further fitted an LMM to test for regional effects of MPC axis loadings on

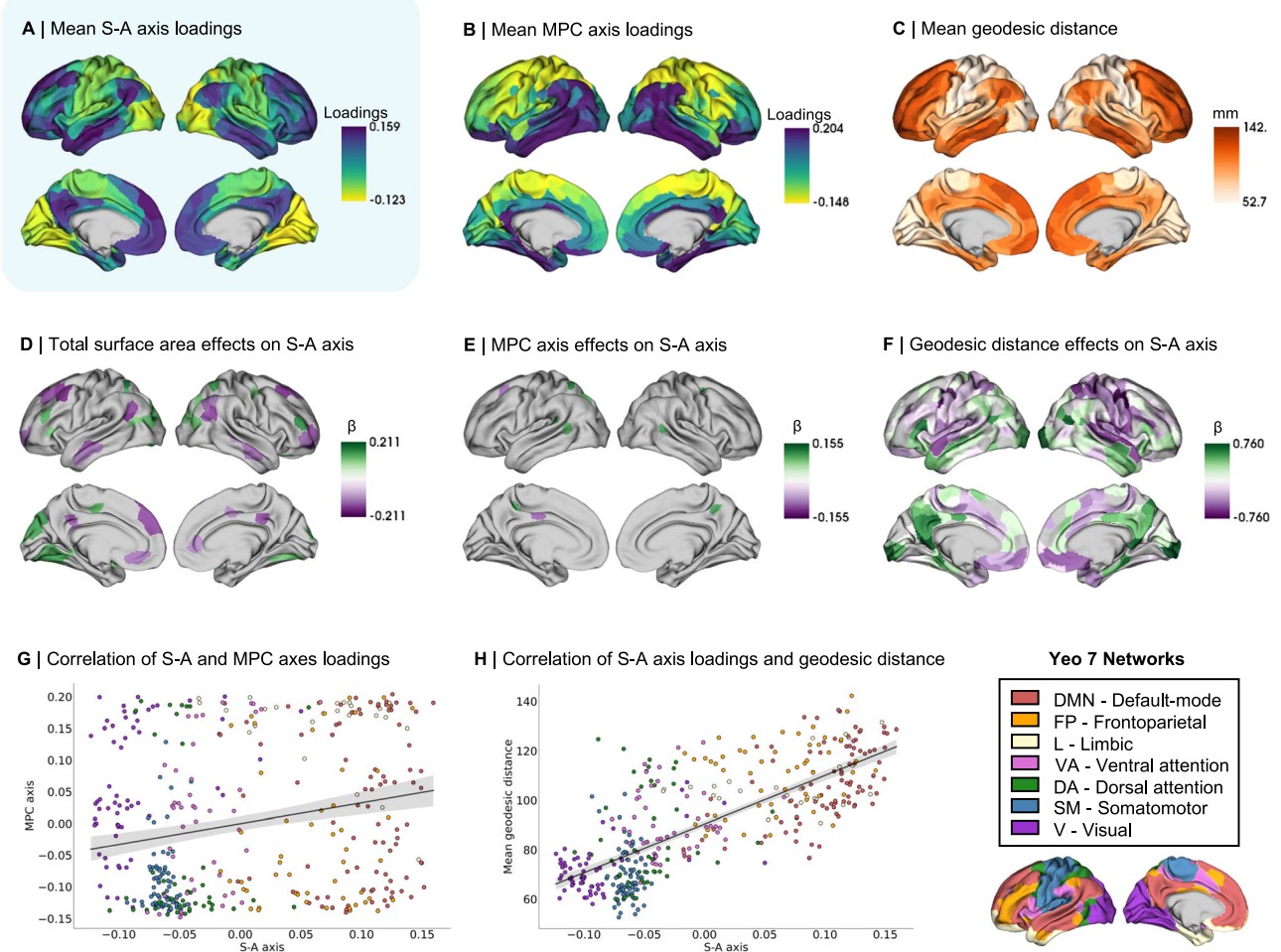

**Fig. 2 | Morphometric correlates of the sensory-association (S-A) axis of functional cortical organization across sexes. A** Mean S-A axis loadings (spanning from visual to default-mode regions) across sexes; **B** Mean microstructural profile covariance (MPC) axis loadings (spanning from sensory to paralimbic regions) across sexes; **C** Mean geodesic distance of connectivity profiles across sexes; **D** Thresholded β-map of linear mixed effect model (LMM) results showing false discovery rate (FDR)-corrected statistically significant effects ($q < 0.05$) of total surface area on the S-A axis loadings; **E** Thresholded β-map of LMM results showing FDR-corrected statistically significant effects of MPC axis loadings on the S-A axis loadings; **F** Thresholded β-map of LMM results showing FDR-corrected statistically

significant effects of mean geodesic distance on the S-A axis loadings; **G** Spatial correlation between mean patterns of S-A axis loadings and mean patterns of MPC axis loadings (color-coded by functional Yeo network), tested by a two-sided Spearman correlation and corrected for spatial autocorrelation, $r = 0.20$, $p_{spin} = 0.037$. Error band displays 95% confidence interval; **H** Spatial correlation between mean patterns of S-A axis loadings and mean patterns of mean geodesic distance (color-coded by Yeo network), tested by a two-sided Spearman correlation and corrected for spatial autocorrelation, $r = 0.76$, $p_{spin} < 0.001$. Error band displays 95% confidence interval. β standardized beta coefficient. Source data are provided as a Source Data file.

S-A axis loadings at the parcel level (Fig. 2E), and found small and localized associations between the S-A and MPC axes.

Third, we computed the mean geodesic distance of connectivity profiles at the individual level. The mean geodesic distance of connectivity profiles is the mean distance along the cortical mantle between each region and its top 10% maximally functionally connected regions. Group-level patterns (i.e., averaged across all subjects; Fig. 2C) revealed shorter geodesic distances in visual and somatomotor (sensory) regions, and longer distances in frontoparietal and DMN (association) regions. We also tested for whole-brain associations between the S-A axis and patterns of mean geodesic distance of connectivity profiles by correlating their spatial maps (Fig. 2A, C) averaged across all subjects (Fig. 2H; $r = 0.76$, $p_{spin} < 0.001$). We also fitted an LMM to test for regional effects of mean geodesic distance on S-A axis loadings at the parcel level (Fig. 2F) and found strong and widespread associations between the S-A axis and mean geodesic distance of functional connectivity profiles.

## Sex differences in morphometric correlates are not associated with sex differences in the S-A axis

After establishing the cortical morphometric correlates of the S-A axis, we probed whether sex differences in cortical morphometry may reflect sex differences in the S-A axis. To this effect, we first computed sex differences in the morphometric correlates of the S-A axis using independent LLMs. Males displayed a larger total surface area ($1947.35 \pm 1542.24$ cm$^2$) relative to females ($1715.07 \pm 1469.42$ cm$^2$), $\beta = 1.21$, $p < .001$, as reported in Supplementary Table 1. We also identified bidirectional statistically significant regional sex differences in MPC axis loadings (Fig. 3B) as well as in the mean geodesic distance of functional connectivity profiles (Fig. 3C).

We then tested whether the identified sex differences in cortical morphometry may be associated with sex differences in the S-A axis. First, we tested whether sex differences in S-A axis loadings were moderated by total surface area. For this, we modeled an interaction term of sex by total surface area on the S-A axis loadings (Fig. 4A) and found no statistically significant effects across cortical regions. In

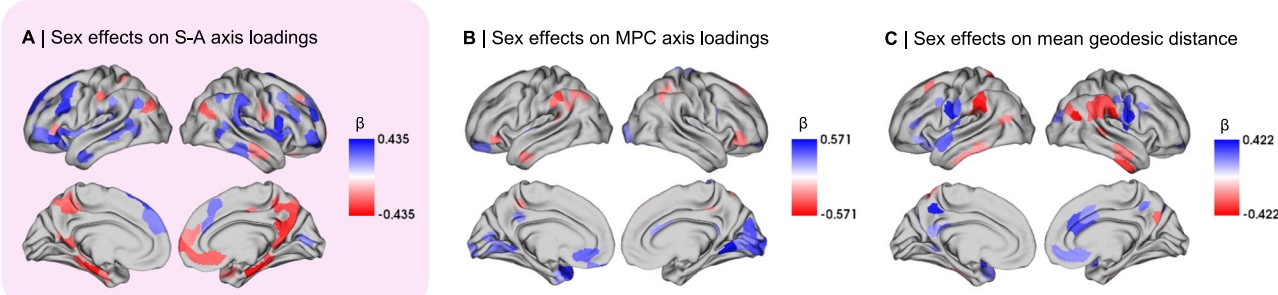

**Fig. 3 | Sex differences in the morphometric correlates of the sensory-association (S-A) axis. A** Thresholded β-map of linear mixed effect model (LMM) results showing false discovery rate (FDR)-corrected statistically significant effects ($q < 0.05$) of sex on the S-A axis, where blue represents higher male loadings and red represents higher female loadings; **B** Thresholded β-map of LMM results showing FDR-corrected statistically significant effects of sex on the microstructure profile covariance (MPC) axis; **C** Thresholded β-map of LMM results showing FDR-corrected statistically significant effects of sex on the mean geodesic distance of connectivity profiles. β standardized beta coefficient. Source data are provided as a Source Data file.

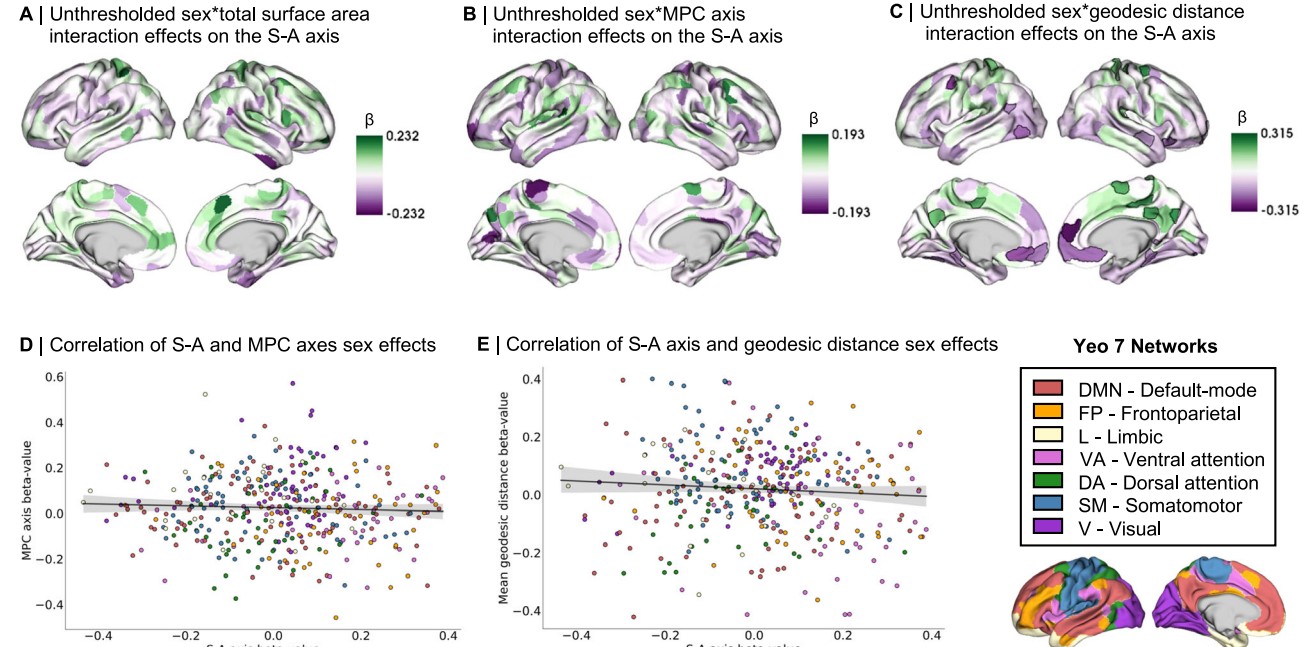

**Fig. 4 | Associations between sex differences in sensory-association (S-A) axis and sex differences in cortical morphometry. A** Unthresholded β-map of linear mixed effect model (LMM) testing for sex by total surface area interaction effects on S-A axis (there were no statistically significant sex by total surface area interaction effects after false discovery rate (FDR) correction ($q < 0.05$)); **B** Unthresholded β-map of LMM testing for sex by MPC axis interaction effects on S-A axis (there were no statistically significant sex by MPC axis interaction effects after FDR correction); **C** Unthresholded β-map of LMM testing for sex by mean geodesic distance interaction effects on S-A axis (35 cortical areas showed statistically significant sex by mean geodesic distance interaction effects after FDR correction and are delineated in black); **D** Scatterplot displaying the spatial correlation between patterns of sex differences (β-maps) in S-A axis loadings and in MPC axis loadings (color-coded by Yeo network), tested by a two-sided Spearman correlation and corrected for spatial autocorrelation, $r = -0.05$, $p_{spin} = 0.398$. Error band displays 95% confidence interval; **E** Scatterplot displaying the spatial correlation between patterns of sex differences (β-maps) in S-A axis loadings and in the mean geodesic distance of connectivity profiles (color-coded by Yeo network), tested by a two-sided Spearman correlation and corrected for spatial autocorrelation, $r = -0.07$, $p_{spin} = 0.326$. Error band displays 95% confidence interval. β standardized beta coefficient. Source data are provided as a Source Data file.

Supplementary Fig. 3A-C, we further show that this interaction effect yields virtually the same effects (β values) when including versus excluding height as a covariate from the LMM ($r = 0.99$, $p_{spin} < 0.001$). This suggests that height—being an anthropometric feature that systematically differs between the sexes—does not explain variance in the moderation of sex effects by total surface area on the S-A axis loadings either. We also plotted within-sex effects of total surface area on S-A axis loadings, showing similar although slightly diverging patterns of effects between males and females ($r = 0.66$, $p_{spin} = 0.002$; Supplementary Fig. 3D–F). However, the divergence of patterns between sexes may not be strong or systematic enough to be interpreted as

meaningful, as underlined by the lack of statistically significant sex by total surface area interaction effects on the S-A axis.

We then tested for spatial associations between regional sex effects on S-A axis loadings (Fig. 3A) and regional sex effects on MPC axis loadings (Fig. 3B) by correlating the two β-maps. Here, we found no statistically significant association between these two patterns of sex differences (Fig. 4D; $r = -0.05$, $p_{spin} = 0.398$). Similarly, we tested for spatial associations between regional sex effects on S-A axis loadings (Fig. 3A) and regional sex effects on the mean geodesic distance of connectivity profiles (Fig. 3C). Again, we found no statistically significant association between these two patterns of sex differences

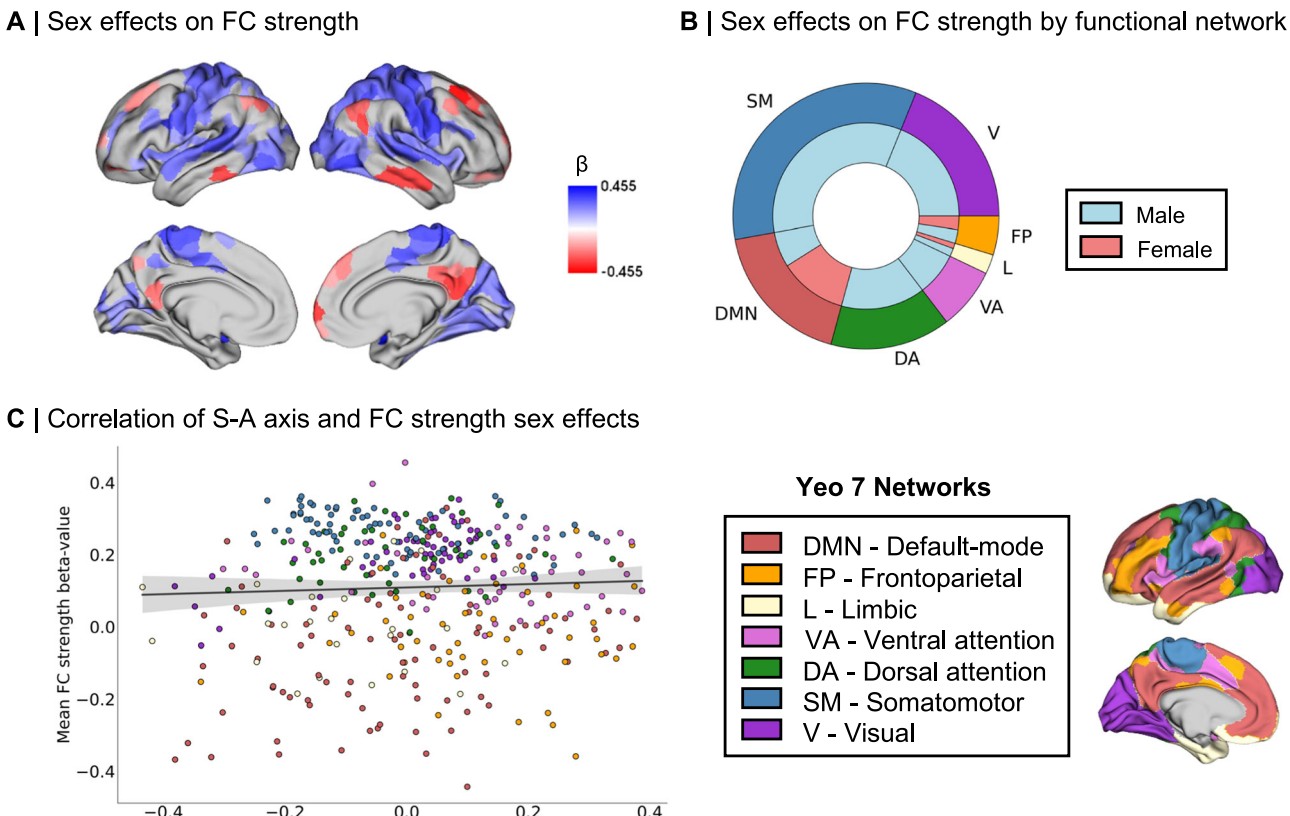

**A | Sex effects on FC strength**

**B | Sex effects on FC strength by functional network**

**C | Correlation of S-A axis and FC strength sex effects**

**Yeo 7 Networks**

- DMN - Default-mode
- FP - Frontoparietal
- L - Limbic
- VA - Ventral attention
- DA - Dorsal attention
- SM - Somatomotor
- V - Visual

**Fig. 5 | Intrinsic sex differences in functional connectivity (FC) strength.**
**A** Thresholded β-map of linear mixed effect model (LMM) results showing false discovery rate (FDR)-corrected statistically significant effects ($q < 0.05$) of sex on mean FC strength; **B** Functional network breakdown of connections showing statistically significant FDR-corrected sex differences in mean FC strength. The outer ring displays absolute proportions of statistically significant cortical areas by functional Yeo network, the inner ring displays absolute nested proportions by directionality of effects, where blue represents greater male FC strength and red represents greater female FC strength; **C** Spatial correlation between patterns of sex effects in S-A axis loadings and patterns of sex effects in mean FC strength (color-coded by Yeo network), tested by a two-sided Spearman correlation and corrected for spatial autocorrelation, $r = -0.02$, $p_{spin} = 0.483$. Error band displays 95% confidence interval. β standardized beta coefficient. Source data are provided as a Source Data file.

(Fig. 4E; $r = -0.07$, $p_{spin} = 0.326$). These results together suggest that sex differences in the S-A axis are not overall systematically associated with sex differences in cortical morphometry. In addition to probing spatial associations between sex differences in cortical morphometry and sex differences in the S-A axis, we also modeled interaction terms of sex by MPC axis loadings (Fig. 4B) and sex by mean geodesic distance (Fig. 4C) to test for these interaction effects on S-A axis loadings. Here, we found no statistically significant sex by MPC axis interaction effects, but find 35 cortical regions showing statistically significant sex by geodesic distance interaction effects, specifically in the bilateral medial prefrontal cortex, temporal regions, and left dorsolateral regions (delineated in black on Fig. 4C).

As an additional sensitivity analysis, we found that including the MPC axis and mean geodesic distance as covariates to an LMM testing for sex effects on the S-A axis yields highly similar regional sex effects to those reported in Fig. 1 (for which the original LMM only included total surface area as a morphometric covariate, to control for brain size), as shown by the strong correlation of β-maps ($r = 0.95$, $p_{spin} < 0.001$, Supplementary Fig. 4A). Similarly, the association between sex effects when including all morphometric covariates versus not including any (i.e., also excluding total surface area) remains high despite a small decrease in correlation strength ($r = 0.80$, $p_{spin} < 0.001$, Supplementary Fig. 4B). These findings further suggest that sex differences in total surface area—representing brain size—only explain a minor amount of variance in sex differences in the S-A axis.

### Intrinsic functional underpinnings of sex differences in the S-A axis
Given that sex differences in the morphometric correlates of the S-A axis did not appear to reflect sex differences in the S-A axis, we probed potential intrinsic functional underpinnings of sex differences in the S-A axis. Concretely, we investigated which sex differences in features of FC may be conserved through data reduction and reflected in the sex differences that we observe in the S-A axis loadings. For this, we tested for associations between sex differences in the S-A axis loadings and sex differences in three intrinsic features of FC, namely mean FC strength, FC profiles, and network topology.

First, we computed mean FC strength at the individual level from FC matrices, representing—for each parcel– the mean row-wise $z$ values of each given seed region's top 10% maximally functionally connected regions. We then fitted an LMM to test for local effects of sex on mean FC strength (Fig. 5A, B), which revealed—amongst other sex differences—greater female intrinsic FC in DMN regions and greater male intrinsic FC in somatomotor regions. To test associations between patterns of sex differences in the S-A axis loadings (Fig. 1B) and in FC strength (Fig. 5A), we spatially correlated the β-maps of the respective sex effects and did not detect a statistically significant association between sex differences in the S-A axis and sex differences in FC strength (Fig. 5C; $r = -0.02$, $p_{spin} = 0.483$).

Second, we investigated whether sex differences in the S-A axis may be related to sex differences in FC profiles. We defined FC profiles at the individual level, for which we identified the top 10% maximally functionally connected regions. Using the Chi-square ($\chi^2$) test of

**A | Seed areas displaying sex differences in their FC profiles**

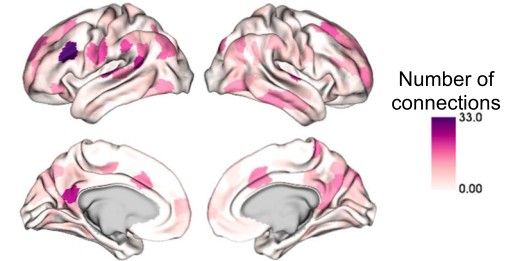

**B | Sex effects on FC profiles by functional network**

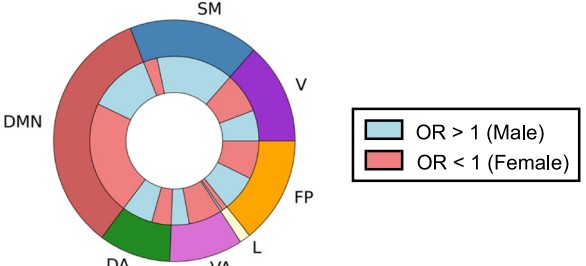

**C | Connections showing sex differences**

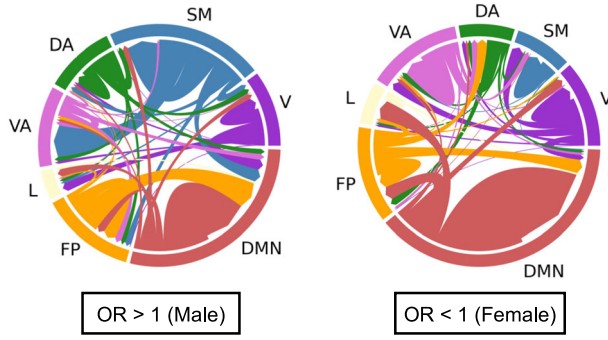

**D | Sex differences in pairwise BN dispersion**

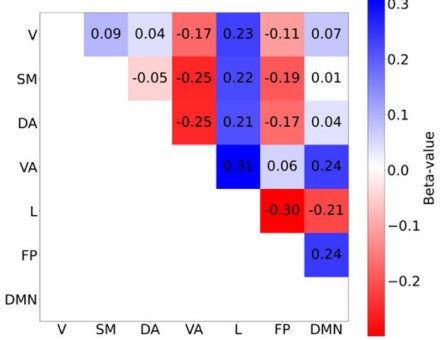

**E | Sex differences in WN dispersion**

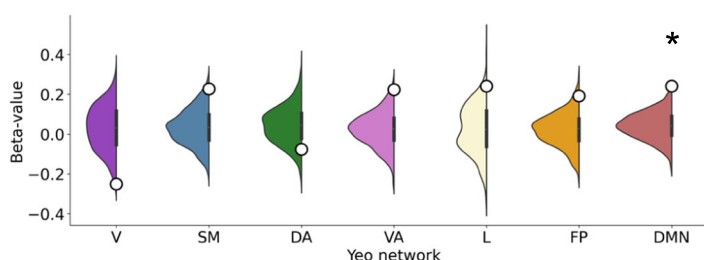

**Yeo 7 Networks**

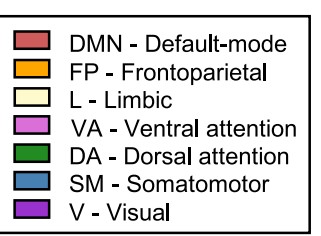
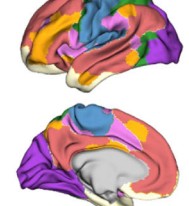

**Fig. 6 | Intrinsic sex differences in functional connectivity (FC) profiles and network topology. A** Number of connections (per seed region) showing statistically significant false discovery rate (FDR)-corrected sex differences in their odds of belonging to the given seed's top 10% connections; **B** Functional network breakdown of connections showing statistically significant FDR-corrected sex effects in their odds of belonging to the given seed's top 10% connections. The outer ring displays absolute proportions of statistically significant cortical areas by functional Yeo network, the inner ring displays absolute nested proportions by directionality of effects, where blue represents higher male odds and red represents higher female odds; **C** Connections between seed and target regions showing statistically significant FDR-corrected sex differences in FC profiles (odds ratio (OR) > 1 meaning that males have higher odds than females of having a target region belong to a seed region's top 10% connections; OR < 1 meaning that females have higher odds than males of having a target region belong to a seed region's top 10% connections; connections are color coded by Yeo network and weighed by number of connections between the network pairs; **D** β values of linear mixed effect model (LMM) results for the sex contrast in between-network (BN) dispersion for each pairwise Yeo network comparison, where blue represents higher male BN dispersion and red represents higher female BN dispersion (no statistically significant sex effects after spin permutation and multiple comparisons Bonferroni correction; two-sided $p_{spin} < 0.001$; **E** β values of LMM results for the sex contrast in within-network (WN) dispersion for each Yeo network (displayed as white dots), plotted on null distributions of β values derived from 1000 spin permutations, where positive β values represent higher male WN dispersion and negative β-values represents higher female WN dispersion. * indicates multiple comparisons Bonferroni-corrected ($p_{spin} < 0.004$) two-sided statistical significance of the sex contrast for dispersion in the DMN, β = 0.241, $p_{spin} < 0.001$. β standardized beta coefficient. Source data are provided as a Source Data file.

independence, we assessed—for each possible pairwise connection along the 400 × 400 matrix—sex differences in a given target region's odds of belonging to the top 10% maximally functionally connected regions of a given seed region. In this way, we specifically considered the data—at the individual level—that our data reduction algorithm has been applied on in order to probe whether the top 10% connections made by females and males may have differentially influenced the computation of the S-A axis and consequently of sex differences in its loadings. We identified the direction of these sex effects with the odds ratio (OR), where OR < 1 indicates a given region's greater female odds and OR > 1 indicates a given region's greater male odds. Out of the 160,000 tested

functional connections, 2004 connections (corresponding to 1.25% of all connections) showed statistically significant sex differences in their odds of constituting a seed's top 10% connections after FDR correction, suggesting that sex differences in S-A axis loadings may in part stem from differences in FC profiles, namely differences in which functional connections are the strongest. For connections showing statistically significant sex differences, we found an OR ranging from 0.00–0.64 in the case of greater female odds (OR < 1), and an OR ranging from 1.56 to 25.36 in the case of greater male odds (OR > 1). For illustrative purposes, we summarized spatial patterns of sex differences in FC profiles as the sum of connections showing sex differences per seed region (Fig. 6A), as

well as the overall networks involved in sex differences in FC profiles (Fig. 6B). Figure 6C displays the connections between seed and target regions showing statistically significant sex differences in their FC profiles.

Finally, we investigated sex differences in network topology, namely the organization of functional networks along the S-A axis. We computed measures of between-network dispersion, quantifying the pairwise distance between all networks along the S-A axis, where a higher value indicates greater segregation of the given pair of networks and a lower value indicates greater integration of the given pair of networks. Here, we computed between-network dispersion for each possible network pair (21 pairs of Yeo networks in total) at the individual level[43]. We also computed a measure of within-network dispersion[40], quantifying the spread of regions within a network along the S-A axis, where a higher value indicates greater segregation of the given network's regions and a lower value indicates greater integration of the given network's regions. Here, we computed within-network dispersion for all seven Yeo networks at the individual level. LMMs did not show any statistically significant sex difference in between-network dispersion for any of the network pairs (Fig. 6D). However, we found greater male within-network dispersion in the DMN, $\beta = 0.241$, $p_{spin} < 0.001$ (Fig. 6E), revealing a greater spread of regions belonging to the DMN along the S-A axis in males. The full statistical results for the analysis of sex differences in network dispersion are summarized in Supplementary Table 2.

## Discussion

In the current work, we investigated the extent to which sex differences in functional cortical organization may be associated with differences in cortical morphometry, namely different measures brain size (focusing on total surface area), microstructure, and the geodesic distance of connectivity profiles. We identified widespread sex differences in young adult functional cortical organization as defined by the S-A axis, which however did not appear to be systematically associated with sex differences in total surface area, microstructural organization, nor the mean geodesic distance of connectivity profiles. This finding is particularly striking given that the morphometric properties under study were all per se associated with the S-A axis and differed between sexes. We observed that sex differences in the S-A axis were instead related to differences in FC profiles and network topology, namely greater male dispersion within the DMN. Collectively, our findings suggest that sex differences in functional cortical organization extend beyond neuroanatomical sex differences pertaining to cortical morphometry.

Considering the common use of different measures of brain size in the literature, which yield different magnitudes of sex differences depending on the selected measure[6], we tested the effects of different measures of brain size on the S-A axis, namely ICV, TCV, and total surface area. Here, given that total surface area had the most widespread effects on functional organization, we deemed it the most appropriate measure of brain size for our study and further included it as a covariate in our models throughout our analyses. The relevance of total surface area for our study is also supported by the theoretical assumptions motivating our research question, namely the relevance of cortical shape and geometry in constraining brain wide functional dynamics[29-31] and thus sex differences in these features potentially underpinning sex differences in the S-A axis. In showing the diverging statistical effects of different measures of brain size, our findings highlight the risk of introducing noise when including an inadequate measure of brain size, particularly when statistically controlling for brain size in the detection of sex effects on brain structure and function[32,44-47]. The complex heterogeneity of neuroanatomical properties constituting brain size should not be undermined, also considering that morphometric features vary differently as a function of age, whereby for example total brain volume but not ICV is affected by

atrophy[6]. As such, future research on sex differences should also carefully select the measure of brain size that is most conceptually and empirically pertinent to the research question under study in order to avoid introducing noise in the analyses.

By establishing morphometric correlates of the S-A axis in addition to total surface area, namely a microstructural axis of cortical organization[37,42,48] and the mean geodesic distance of connectivity profiles[10,48], our findings align with previous work and argue for the rooting of functional cortical organization in cortical structure and shape. We show a particularly strong association between the mean geodesic distance of connectivity profiles and the S-A axis, supporting the relevance of the cortical mantle's shape in sculpting functional organization. This may be a product of the cortical mantle's evolutionary expansion, where association regions are untethered from sensory hierarchies[13], and long-range connections preserve the overall connectedness of cortical networks by facilitating the communication between distant areas[28]. Furthermore, as indexed by the MPC axis, microstructural organization appears to mildly covary with the S-A axis, supporting to some degree the well-established idea of structural constraints on brain function[37,42,49]. In our study, we obtained intensity profiles via the ratio of T1w over T2w imaging sequences, and although it is commonly used to measure myelin[37,42,50], the T1w/T2w ratio has been described as an acceptable *qualitative* proxy for myelin in gray but not white matter[51]. It is indeed thought to capture unique features of microstructural tissue that appear largely independent of diffusion-based metrics, thus portraying a mix of neuroanatomical features beyond pure myelin[52]. We therefore consider the T1w/T2w ratio—and the resulting MPC axis—as a general measure of tissue microstructure, which may serve as a scaffold for functional organization.

After establishing morphometric correlates of the S-A axis, we addressed our primary aim of probing the extent to which sex differences in functional cortical organization may be reflected by sex differences in cortical morphometry. Our findings overall suggest that morphometric differences between the sexes are altogether not substantial contributors to sex differences in the S-A axis of functional organization. We did not find any statistical spatial associations between patterns of sex differences in the S-A axis and patterns of sex differences in the MPC axis nor in the mean geodesic distance of connectivity profiles. Although we observed slightly diverging results when including—as opposed to excluding—total surface area as a covariate in our model testing for sex differences in S-A axis loadings, we did not find a statistically significant interaction between sex and total surface area in reflecting S-A axis loadings. We also found no statistically significant sex by MPC axis interactions but found few statistically significant sex by mean geodesic distance interactions. This can be understood as partly mirroring the sex differences in functional connectivity profiles that we further explain below, given that geodesic distance was averaged based on the connectivity profiles—more specifically the top 10% functional connections—which we find sex differences in. Altogether, the negligeable relevance of cortical morphometry to sex differences in the S-A axis is striking given that morphometric properties appear per se to be associated with the S-A axis and to differ between sexes. The biological mechanisms underpinning different patterns of morphometric and functional sex differences may thus be independent from one another, suggesting that sex differences in functional cortical organization may extend beyond the connectome's supporting shape and structure.

Given that sex differences in morphometric correlates of the S-A axis did not seem to be associated with sex differences in the S-A axis, we probed and found potential intrinsic functional underpinnings of sex differences in the S-A axis. Firstly, the sex differences that we observed in the S-A axis loadings were distributed across functional networks, notably in the DMN, frontoparietal, and ventral attention networks. This is consistent with a previous study in youth reporting that these association networks show greater individual variability in

their functional topography relative to lower-order sensory networks, whilst contributing the most to sex classification[8]. We also observed sex differences in intrinsic FC strength, replicating previous widely established findings of greater FC in females within DMN regions[21–23] and in males within somatomotor regions[22,25]. However, these patterns did not spatially overlap with patterns of sex differences in the S-A axis, suggesting that FC strength is not a feature of intrinsic FC that is captured by sex differences in our low dimensional representation of functional organization. Instead, we found that sex differences in the S-A axis were related to modest sex differences in FC profiles, which also presented qualitative sex differences in the proportional breakdown of networks involved. The strongest functional connections (top 10% connections) of females seemed to involve the DMN more than in males, whereas males displayed more top connections involving the somatomotor networks relative to females. This is also consistent with the few regional statistically significant interaction effects of sex by mean geodesic distance of FC profiles on S-A axis loadings. Indeed, given that sensory and association regions are respectively known to have primarily short- versus a mix of short- and long-range connections[26], sex differences in FC profiles involving regions in the somatomotor network and the DMN may be related to differences in the geodesic distances of functional connectivity profiles between sexes. As such, our findings suggest that sex differences in the S-A axis may be better represented by sex differences in FC profiles, i.e., the topology of functional connections, than sex differences in FC strength alone. Furthermore, sex differences in the configuration of functional connections may not only underly the recurrence of sex differences in these networks[21–23,25], but may also reflect sex differences in network topology.

We observed greater male dispersion relative to females within the DMN. This finding suggests that areas belonging to the DMN are represented further apart on the S-A axis (i.e., showing less similarity in their FC profiles) in males relative to females, which is also consistent with previous findings of generally more segregated male networks[53]. These network-specific topological sex differences may be related to greater female odds of connections within the DMN, and greater male odds of somatomotor connections with other networks. As such, the apparent sex differences in network topology and functional connectivity profiles—albeit small—provide a more interpretable system-level description of the sex differences observed in the S-A axis loadings, representing a key principle of macroscale cortical organization. Concretely, network topology, which represents the organization of functional communities within and between functional networks[43], may reflect brain states[54]—and in our case, possible differences thereof at the group-level. Network topology has also been associated with different cognitive features including arousal[55], awareness and consciousness[56], behavior and task performance[57], and cognitive flexibility[58]. The balance between integration and segregation is complex, dynamic, and necessary to maintain the brain's metastability[59] by reaching a point of equilibrium between global organization and local specialization[49]. The brain is a highly interconnected and metabolically expensive organ, and its organization is required to dynamically balance topological efficiency and energy utilization in response to transient cognitive and physiological demands[60]. Our findings of sex differences in network topology may therefore pertain to intricate sex differences not only in group-averaged brain states at rest, but also in global energy expenditure, which would reflect physiological differences.

Despite the insights gained through our study, some limitations must be acknowledged. Firstly, by focusing on self-reported biological sex, we did not test for effects of gender or gender-related variables on functional organization and its morphometric correlates. This is relevant given that there are multiple factors contributing to sex/gender differences, including both biological and social factors. Findings may indeed appear more nuanced if we move

beyond the unrealistic assumption of a clear-cut sexual dimorphism of brain structure and function[61], further considering the relevance of gender, steroid hormones, and the role of X and Y chromosomes[62]. Nevertheless, we intentionally focused on the dichotomous variable of sex to identify group-level effects, as our study aimed to investigate the correspondence (i.e., shared variance) between sex differences in cortical morphometry and sex differences in cortical functional organization, regardless of variance explained by gender. We recognize the limitations of using a binary variable given that differences between groups may be attributable to both sex- and gender-based influences[62]. However, quantifying these influences was beyond the scope of this study. Secondly, we focused on neocortical functional organization, excluding subcortical structures and the cerebellum despite their substantial contributions to whole brain organization through their notable structural integration with the cortex[63]. For example, the amygdala and hippocampus have shown a variable degree of both structural[6] and functional[64] sex differences. Nevertheless, our exclusive focus on the neocortex was motivated by the relevance of using the S-A axis as our measure of functional organization, which is commonly obtained by reducing the dimensionality of FC matrices of cortical data[10]. By using the S-A axis, our work took a system-level approach to identify sex differences embedded in a key macroscale organizational principle, going beyond previous research on functional differences between the sexes solely focusing on intrinsic brain function. Thirdly, the morphometric properties considered in our study are not exhaustive, overlooking the contributions of other morphometric measures such as local volumes of gray matter. The inclusion of the MPC axis[37,42] and the mean geodesic distance of connectivity profiles[10,48] was however supported by their theoretical and empirical relevance to functional cortical organization, particularly its low dimensional embedding. Finally, our findings are limited to sex differences in a healthy sample and would benefit from being replicated in a more inclusive sample that is more representative of the overall population. This would be additionally informative considering the notable sex differences observed in populations that would typically be excluded from healthy samples, for example individuals with neurodevelopmental and psychiatric disorders[65]. Nevertheless, the structure-function associations that we investigated in this work are rather fundamental, and their essence should thus be fairly well captured in our large (healthy) sample.

All in all, our study gives rise to a set of questions pertaining to the mechanisms underpinning sex differences in functional cortical organization, given that they do not appear to be rooted in cortical morphometric differences. Our findings instead suggest that sex differences in the S-A axis are, to some extent, intrinsically related to differences in FC profiles and network topology. Although these sex differences appear to be small, they may be meaningful for broader sex differences in functional cortical organization, and future research should explore factors driving males and females to form these few distinct functional connections that are associated with sex differences in the system-level organization of functional networks, notably of the DMN. Recognizing the human body as a complex system of systems, future work should also investigate other biological factors that may contribute to functional sex differences such as genes located on sex chromosomes[19] and steroid hormones[66,67]. Environmental factors should equally be considered, as they may not only differ on average as a function of sex and gender, but may also differently affect brain function across the sexes through divergent mechanisms[68]. An example of this is stress, whereby sex differences in the stress response have been found to contribute to sex differences in brain function and psychopathology via epigenetic mechanisms[69–71]. Generally, it is crucial for research in neuroscience to systematically test for sex differences in brain structure and function, as well as their biological and environmental underpinnings, in order to produce more rigorous and

representative findings, ultimately leading to a more translational body of knowledge[68].

## Methods

The current study complies with ethical regulations set by The Independent Research Ethics Committee at the Medical Faculty of the Heinrich-Heine-University of Duesseldorf.

### Participants and study design

Our analyses were conducted on the publicly available data of healthy young adults from the Human Connectome Project (HCP) S1200 release (http://www.humanconnectome.org/)[33]. We selected subjects with available functional, T1, and T2 neuroimaging data, resulting in a final sample of 1000 individuals (536 females) with a mean age of $28.73 \pm 3.71$ years, used for all analyses. The sample included 284 monozygotic twins (MZ), 184 dizygotic twins (DZ), 443 non-twin siblings, and 89 unrelated individuals, and the sociodemographic breakdown of the participants is additionally reported in Supplementary Table 3. Subjects were all born in Missouri but recruited in an attempt to broadly reflect the racial and ethnic composition of the United States population. Recruitment efforts aimed to yield a subject pool capturing a wide range of variability—in socioeconomic and behavioral terms—in order to be representative of the general healthy population. The term "healthy" was thus broadly defined. Individuals with documented neurodevelopmental and psychiatric disorders, or reporting physiological illnesses such as high blood pressure or diabetes were excluded from the HCP study recruitment protocol, but not individuals who reported smoking, being overweight, or a history of recreational drug use or heavy drinking (if they had not experienced severe symptoms). Informed consent was obtained for all study subjects and recruitment procedures were approved by the Washington University institutional review board. More detailed information about the HCP study design and recruitment procedure is available elsewhere[33,72]. Despite the use of the term *gender* in the HCP Data Dictionary, we use the term *sex* in this article given that the HCP study collected self-reported information on biological sex instead of gender identification, as reported elsewhere[73]. We have not used genetic information to verify the self-reported sex.

### Structural MRI acquisition and preprocessing

The HCP's MRI data was acquired on a customized 3T Siemens Skyra ConnectomeScanner with a 32-channel head coil at Washington University across four scanning sessions held over two days. Structural MRI images were acquired on the same day via high resolution T1-weighted (T1w) and T2-weighted (T2w) sequences. Two separate T1w images were acquired and averaged, with identical scanning parameters using a 3D MPRAGE sequence (0.7 mm isovoxels, FOV = 224 mm, matrix = 320 × 320 mm, 256 sagittal slices; TR = 2400 ms, TE = 2.14 ms, TI = 1000 ms, flip angle = 8°, BW = 210 Hz per pixel, ES = 7.6 ms). Two separate T2w images were acquired and averaged, with identical scanning parameters using a variable flip angle turbo spin-echo (3D T2-SPACE) sequence, with the same isotropic resolution, matrix, FOV, and slices as for the T1w sequence (TR = 3200 ms, TE = 565 ms, BW = 744 Hz per pixel, total turbo factor = 314). The preprocessing steps included co-registering the T1w and T2w images, bias field (B1) correction, registration to MNI space, segmentation, and surface reconstruction. See refs. 33,72,74 for more detail on the HCP's MRI protocols and the FreeSurfer segmentation pipeline.

### Functional MRI (fMRI) acquisition and preprocessing

The HCP's fMRI data was collected after the structural sequences and following the HCP's minimal processing pipeline, as described above. A total of 1 h of resting-state functional data was collected across four identical 15 min scanning sessions, equally split over two days (LR1, RL1, LR2, RL2), with a gradient echo EPI sequence at a resolution of 2 mm isotropic (FOV = 208 × 180 mm, matrix = 104 × 90 mm, 72 slices covering the whole brain, TR = 720 ms, TE = 33 ms, multiband factor of 8, FA = 52°). The multimodal surface matching algorithm (MSMAll) was used to co-register the data to the HCP template 32k_LR surface space, consisting of 32,492 nodes per hemisphere (59,412 nodes excluding the medial wall). A more detailed description of the resting state fMRI data acquisition and analysis protocol is available elsewhere[74,75].

### Functional connectivity (FC) and the sensory-association (S-A) axis of functional organization

Throughout this work, we used the Schaefer 400 parcellation (clustered into 7 networks: visual, somatomotor, dorsal attention, ventral attention, limbic, frontoparietal, DMN[40]). This widely used functionally-derived parcellation scheme was originally obtained via a gradient-weighted Markov Random Field model integrating local gradient and global similarity approaches[76]. The vertex-wise functional timeseries were therefore averaged within the Schaefer 400 cortical regions. FC matrices (400 × 400) were then computed at the individual level—per scanning session—by correlating cortical timeseries in a pairwise manner using the Pearson product moment. We normalized the correlation coefficients using Fisher's z-transformation. Final FC matrices were obtained by averaging each subject's matrices across their four scanning sessions. From these FC matrices and for each subject, we computed the S-A axis of functional organization, as described below.

We conducted data reduction on the FC matrices to yield macroscale gradients of functional organization[10]. For this, we used diffusion map embedding, a nonlinear manifold learning algorithm that reduces complex, high-dimensional structures of data (in our case affinity matrices) to low-dimensional representations combining geometry with the probability distribution of data points[34]. Thus, cortical regions that are strongly interconnected (i.e., whose timeseries show high correlations) are represented closer together in the resulting low dimensional manifold of FC data, whereas parcels with low covariance are represented farther apart, as indexed by the cortical regions' gradient loadings. To this end, we used the BrainSpace Python toolbox[35] to compute ten gradients with the following parameters: 90% threshold (i.e., only considering the top 10% row-wise z values of FC matrices, representing each seed region's top 10% maximally functionally connected regions), $\alpha = 0.5$ ($\alpha$ controls whether the geometry of the set is reflected in the low-dimensional embedding— i.e., the influence of the sampling points density on the manifold, where $\alpha = 0$ (maximal influence) and $\alpha = 1$ (no influence)), and $t = 0$ (t controls the scale of eigenvalues). These parameters were selected for consistency with previous studies and represent choices that are recommended to retain global relations between datapoints in the embedded space whilst being relatively robust to noise[10,42]. To confirm the robustness of the S-A axis computed at the 90% threshold, we further show with a sensitivity analysis that the mean S-A axis computed at the 90% threshold shows high correlations with mean S-A axes computed at different thresholds (from 10% to 90%, in steps of 10%), with r values ranging from 0.84 to 0.93 (see Supplementary Fig. 5). In order to increase comparability for further between-subject analyses, Procrustes alignment was used to align individual gradients to mean gradients. Mean gradients were computed by applying diffusion map embedding—with the same parameters listed above—to the mean FC matrix (i.e., FC matrices averaged across all subjects). The computation of these FC gradients was carried out independently per hemisphere (i.e., considering the top 10% row-wise z values of only half of the FC matrices, shaped 200 × 200) and the gradient loadings resulting from each hemisphere were subsequently concatenated. This decision was made for consistency and comparability reasons within our study, so that the top 10% functional connections selected for data reduction corresponded to those considered in the calculation of the mean geodesic distance of connectivity profiles—which were only

computed per hemisphere– as described further below). We verified and confirmed the stability FC gradients when computing them per hemisphere versus at the whole brain level, as shown by the spatial correlation of mean gradient loadings ($r = 0.98$, $p_{spin} < 0.001$). Finally, we took the well-replicated principal gradient explaining the most variance in the data and spanning from visual to DMN regions[10], which we labeled the S-A axis and used to represent functional organization for subsequent analyses.

We also computed, for each subject, mean FC strength at the parcel level in a seed-wise fashion, by averaging the row-wise $z$ values of each seed region's top 10% maximally functionally connected regions–again per hemisphere–and subsequently concatenated the hemispheric mean FC strength values to reconstruct whole brain data.

### Cortical microstructure and microstructural profile covariance (MPC)

Microstructural properties–including myelin and cellular characteristics–show depth-dependent variation along cortical columns, as reported by histology[42,77,78] as well as in vivo and post mortem neuroimaging[37,41,42,78], which illustrate a cortical hierarchy[11]. Similar to previous work[37], we quantified cortical microstructure, or "microstructural profile intensity" (MPI), using the myelin-sensitive MRI contrast obtained from the T1w/T2w ratio from the HCP minimal processing pipeline described above[74] (a reliability check is reported in the Supplementary Fig. 6). The T1w/T2w ratio uses the T2w image to correct for inhomogeneities in the T1w image[50]. Then, we followed the previously described protocol[37,41,42] to compute our measurement of MPC, which reflects the variation of MPI, across cortical depths. In short, we generated 14 equivolumetric surfaces within the inner and outer cortical surfaces, then excluded the inner- and outer-most surfaces, thus remaining with 12 surfaces representing cortical layers. Surface generation was based on a model compensating for cortical folding by altering the pairwise Euclidean distance ($\rho$) of intracortical surfaces throughout the cortex and thus preserving fractional volume between the surfaces. For each surface, $\rho$ was calculated as defined in Eq. 1.

$$\rho = \frac{1}{A_{out} - A_{in}} \cdot \left( -A_{in} + \sqrt{\alpha A_{out}^2 + (1-\alpha)A_{in}^2} \right) \qquad (1)$$

for which $\alpha$ denotes a fraction of the total volume of the segment that the surface accounts for, while $A_{out}$ and $A_{in}$ respectively denote the surface areas of the outer and inner cortical surfaces.

Across the whole cortex and from the outer to the inner surfaces, we systematically sampled MPI values layer-wise for each of the 64,984 vertices of the HCP template 32k_LR surface space, which we then averaged within each of the 400 Schaefer cortical areas, per layer. Following a previously described protocol[41], we constructed subject level 400 × 400 matrices using pairwise Pearson partial correlation on the MPI profiles of cortical parcels (i.e., correlating the MPI values across 12 layers between parcels), controlling for overall mean cortical MPI, followed by log transformation. We then used these matrices to compute MPC gradients by following the same procedure and using the same toolbox and parameters as for computing the FC gradients[34,35], as described above and previously done[37,41,42]. For consistency and comparability with the computation of FC gradients and mean geodesic distance of connectivity profiles, we also computed MPC gradients independently per hemisphere and subsequently concatenated the gradient loadings resulting from each hemisphere. We verified and confirmed the stability MPC gradients when computing them per hemisphere versus at the whole brain level, as shown by the spatial correlation of mean gradient loadings ($r = 0.99$, $p_{spin} < 0.001$). We also selected the principal gradient of MPC explaining the most variance in the data and spanning from sensory to paralimbic regions,

which we labeled the MPC axis and used to represent microstructural organization in subsequent analyses.

### Measures of brain size

In our analyses we included different measures of brain size typically used in the literature, including intracranial volume (ICV), total cortical volume (TCV), and total surface area. For ICV, we used the FreeSurfer output measure IntraCranialVol, which is an estimate of ICV based on the Talairach transform. We computed our own measure of TCV by summing the volumes of the TotCort_GM_Vol and Tot_WM_Vol Free-Surfer output measures, which we considered relevant to our study's focus on *cortical* functional organization (thus excluding the volumes of subcortical structures). We computed total surface area by using the FreeSurfer mri_surf2surf tool to resample cortical gray matter surface for each subject.

### Geodesic distance of connectivity profiles

Geodesic distances, representing the shortest distance between two vertices along the folded cortical mantle's curvature, were computed using the Micapipe toolbox[79], and following the previously described protocol[80]. In short, geodesic distance matrices were computed for each subject along their native cortical midsurface. The first step consisted in defining a centroid vertex for each cortical parcel, identified as the vertex having the shortest summed Euclidean distance from all other vertices within the parcel. Then, Dijkstra's algorithm[81] was used to compute geodesic distances between the centroid vertices and all other vertices on the on the native midusrface mesh. The vertex-wise geodesic distance values were then averaged within each parcel to form the geodesic distance matrices. From these individual matrices, we finally averaged–parcel-wise–the geodesic distance values of each seed cortical region's top 10% maximally functionally connected regions per hemisphere, thus obtaining for each subject the mean geodesic distance of functional connectivity profiles by region.

### Statistical analysis

Given that the HCP sample includes different levels of kinship, we used linear mixed effects models (LMMs) to account for sibling status (MZ, DZ, non-twin siblings) and family relatedness. In fact, all LMMs mentioned in this work consistently included sex, age, and total surface area as covariates (unless otherwise mentioned), and controlled for random nested effects of family relatedness and sibling status. In addition, effects on cortical data obtained via LMMs underwent false discovery rate (FDR) correction ($q < 0.05$), thus correcting for multiple comparisons across the 400 Schaefer cortical regions. Throughout this work, we also tested for associations in brain-wide patterns displayed in the form of cortical maps, for which we used Spearman-rank correlation followed by spin-permutation tests to control for spatial autocorrelation[82]. All statistical tests were two-sided.

After computing the S-A axis of functional cortical organization, we tested for sex differences in the S-A axis loadings with an LMM. Then, we investigated which measure of brain size (out of ICV, TCV, and total surface area) had the largest effects on the S-A axis loadings using separate LMMs (respectively only including ICV, TCV, or total surface area as a covariate, in addition to sex, age and the random nested effect of family relatedness and sibling status). The reason underlying our decision to systematically include total surface area as a covariate in all our LMMs (as the measure of brain size) is that it showed the most widespread effects on the S-A axis loadings out of the three tested measures. Then, we investigated associations between the S-A axis and cortical morphometry, namely the MPC axis and the mean geodesic distance of connectivity profiles, using both LMMs and Spearman-rank spatial correlations of cortical maps.

To probe whether sex differences in cortical morphometry may be associated with sex differences in the S-A axis, we tested whether sex differences in the S-A axis loadings were moderated by total surface

**Table 1 | Contingency matrix structure for the computation of sex differences in functional connectivity profiles**

| | Cortical region belongs to the seed region's top 10% maximally functionally connected regions | Cortical region does not belong to the seed region's top 10% maximally functionally connected regions |
|---|---|---|
| Males | Cm | NCm |
| Females | Cf | NCf |

Cm and Cf respectively denote the number of males and females for which the given cortical region (corresponding to the matrix column) constitutes the given seed region's (corresponding to the matrix row) top 10% maximally functionally connected regions; NCm and NCf, respectively denote the number of males and females for which the given cortical region does not constitute the given seed region's top 10% maximally functionally connected regions.

area by modeling an additional interaction term of sex by total surface area on the S-A axis loadings within the original LMM. We also tested for sex differences in the MPC axis and in the mean geodesic distance of connectivity profiles, and conducted Spearman-rank correlations of cortical β-maps for the sex contrast in the S-A axis and in the morphometric measures. We also modeled two additional interaction terms within the original LMMs of sex by MPC axis loadings and sex by mean geodesic distance to show their effects on the S-A axis loadings. Finally, we conducted sensitivity analyses to test for sex effects on the S-A axis yielded by an LMM including all morphometric measures as covariates (i.e., including the MPC axis and the mean geodesic distance of connectivity profiles, in addition to total surface area), as well as an LMM not including any morphometric measures as covariates (i.e., also excluding total surface area). We then tested the similarity of both these sex effects with the original sex effects on the S-A axis with a Spearman-rank spatial correlation of the cortical β-maps.

In order to probe the potential intrinsic functional underpinnings of sex differences in the S-A axis, we tested for sex differences in FC strength (also with an LMM), as well as sex differences in FC profiles, i.e., the presence of sex differences in the top 10% maximally functionally connected regions used to compute the S-A axis. To this end, we built 400 × 400 binary matrices at the subject level—based on the subjects' individual FC matrix z values—in which we marked in a seed-wise fashion (along the matrix rows) whether the given cortical region (along the matrix column) belongs to the given seed's top 10% maximally functionally connected regions, where 1 indicated that the parcel belongs to the seed's top 10% maximally functionally connected regions and 0 indicated that the parcel does not belong to the seed's top 10% maximally functionally connected regions. We then summed the binary matrices separately within sexes in order to fill 160,000 contingency matrices—one for each cell (i.e., functional connection) of the 400 × 400 FC matrix—denoting the number of males and females for which a given cortical region belongs or does not belong to the seed region's top 10% maximally functionally connected regions (see Table 1 for a visual representation of the contingency matrix structure).

We then conducted the Chi-square ($\chi^2$) test of independence (degrees of freedom = 1) on each contingency table to test for sex differences in the odds of each parcel of belonging to the top 10% maximally functionally connected regions of each seed region. Given the large number of tests conducted here (400 × 400 = 160,000), we controlled for multiple comparisons using FDR correction. We quantified the size of these sex effects with the odds ratio (OR), calculated as defined in Eq. 2.

$$OR = \frac{Cm/NCm}{Cf/NCf} \quad (2)$$

where OR > 1 indicates greater male odds—and OR < 1 indicates greater female odds—of a given region of belonging to a given seed's top 10% maximally functionally connected regions.

We also tested for sex differences in network topology, i.e., how nodes are physically organized in networks and how networks are physically organized along the S-A axis. For this, we computed two measures of network dispersion: between- and within-network dispersion. Between-network dispersion is defined as the Euclidean distance between a pair of network centroids, where a higher value indicates that networks are more segregated from one another along the S-A axis. Within-network dispersion is defined as the sum squared Euclidean distance of network nodes (i.e., S-A axis regional loadings) to the network centroid, where a higher value indicates wider distribution and segregation of a given network's nodes along the S-A axis. At the individual level, we thus computed between-network dispersion between all networks in a pairwise fashion (21 pairs), and within-network dispersion for all 7 networks, by defining network centroids as the median of the S-A axis loadings of all parcels belonging to a given network, following a previously described method[43]. Then, we computed sex differences in each of the 21 between-network dispersion metrics and 7 within-network dispersion metrics using LMMs. For each model, we computed a null distribution of β coefficients for sex differences using 1000 spherical rotations of the Schaefer parcellation scheme in order to shuffle the network labels[82], against which we computed our p-value to determine statistical significance. We then assed $p_{spin}$ values against Bonferroni-corrected two-tailed α-levels of 0.001 (0.025/21) and 0.004 (0.025/7) for between-network and within-network dispersion sex contrasts, respectively.

### Reporting summary
Further information on research design is available in the Nature Portfolio Reporting Summary linked to this article.

## Data availability
All data needed to evaluate the conclusions of the paper are present in the paper and in the Supplementary Materials, and are further available upon request. We obtained all data from the open-access Human Connectome Project (HCP) S1200 young adult sample. The HCP processed data are publicly available and can be directly downloaded at https://db.humanconnectome.org. Source data are provided with this paper.

## Code availability
Analyses were conducted in Python and R: The code used in this manuscript is available at https://github.com/biancaserio/sex_diff_gradients (v1; https://zenodo.org/doi/10.5281/zenodo.12785462). The code and tutorials for functional gradient decomposition and to generate geodesic distances can further be found at https://brainspace.readthedocs.io/en/latest/index.html and https://micapipe.readthedocs.io/en/latest/ respectively.

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

## Acknowledgements

We want to thank the Human Connectome Project, Washington University, the University of Minnesota, and Oxford University Consortium (Principal Investigators: David Van Essen and Kamil Ugurbil; 1U54MH091657) originally funded by the 16 N.I.H. Institutes and Centers that support the N.I.H. Blueprint for Neuroscience Research; and by the McDonnell Center for Systems Neuroscience at Washington University. B.S., M.D.H., and G.B. were funded by the German Federal Ministry of Education and Research (BMBF) and the Max Planck Society. J.S. was funded by the Max Planck Society and University of Leipzig. L.W., S.W., and S.B.E. were funded by the European Union's Horizon 2020 Research and Innovation Program (grant agreements 945539 [HBP SGA3], 826421 [VBC], and 101058516), the DFG (SFB 1451 and IRTG 2150), and the National Institute of Health (NIH; R01 MH074457). S.L.V. was supported by the Max Planck Society through the Otto Hahn Award.

## Author contributions

Conceptualization: B.S. and S.L.V. Main analysis and visualization: B.S. Input on analysis: M.D.H., G.B., and S.L.V. Writing—original draft: B.S. Writing—review and editing: B.S., M.D.H., L.W., G.B., J.S., S.W., S.B.E., S.L.V. Supervision: S.L.V.

## Funding

## Competing interests
The authors declare no competing interests.
