## [Peer Review File · Nature Communications]

Sex differences in functional cortical organization reflect differences in network topology rather than cortical morphometryReviewer #1 (Remarks to the Author):

In this manuscript, Serio and colleagues investigated whether sex differences in cortical morphometry may explain sex differences in functional organization in the brain. Their analyses reveal negative findings, and instead suggest that differences in the sensory-association functional gradient between males and females are related to sex differences in functional connectivity. Major strengths of the study include the use of a large dataset and the consideration of multiple morphometric properties in the analyses. The manuscript reads well and the reported results are likely to be of broad interest to the neuroscientific community. Please find below my comments/questions on the work.

1. Why was a 90% threshold used (i.e., only top 10% of functional connections considered) for the analyses? Do these results hold under other thresholds? Demonstrating that these results hold under multiple threshold (or at least yield the same (or similar) gradients) would strengthen the results.
2. The use of the acronyms S-A and SA is confusing, please consider using just one of the two.
3. Figure S2 is missing a label for the colorbar.
4. Table S1 - I believe there may be a typo in the total SA values listed (mean for males is shown as 1947.35 and mean for females is shown as 17150.67).

Reviewer #2 (Remarks to the Author):

In this paper, the authors thoroughly investigate the possible structural and functional underpinnings of sex differences in the sensory-association axis, which can be considered one of the major organizational properties of the functional connectome in humans. They find that the examined structural properties are associated with the S-A axis but do not appear to underpin sex differences in the S-A axis, despite exhibiting sex differences themselves. They additionally find sex differences in other functional connectivity properties, which may relate to sex differences in the S-A axis. The manuscript is clearly written, and the analyses are nicely rigorous. As a whole, I consider this manuscript to be an important contribution to the field that bolsters our understanding of the basic underlying organizational properties of the human brain.

My main comments are as follows:

- The primary finding of the manuscript – that sex differences in S-A are not related to morphometric properties – is informative but also essentially a null result. It would therefore be helpful if the authors reframed that finding as what effect size they had power to detect. This would be similar to when genetics papers find no significant associations between their trait of interest and SNPs; instead of saying there is fundamentally association, they say that – if there is an association – it is no larger than (e.g., medium or small) effect size.
- One of the main findings is that sex differences in the S-A axis are associated with sex differences in functional connectivity profiles. For this analysis the authors assessed sex differences in pairwise functional connectivity (in the top 10% of connections) and interpret these sex differences as partially underpinning sex differences in S-A loadings. However, given that the S-A loadings derive from pairwise functional connectivity (in the top 10% of

connections), the manuscript would benefit from a clearer description of how these findings are not at least partially circular.

- The term 'brain size' is used in an unclear and potentially misleading manner in some sections of the manuscript. Specifically, I would argue that most readers would assume 'brain size' is referring to ICV or TBV if no additional specification is provided. However, in the abstract and elsewhere 'brain size' is used without additional specification to refer to total SA. Similarly, 'TBV' is used to refer to total cortical volume, not total brain volume; this could also be considered misleading. I understand why the authors chose to examine total SA and total cortical volume, but the terminology used throughout the abstract and manuscript should reflect what was examined. If the authors wish to discuss 'brain size' in the abstract without additional specification, then they should complete supplementary sex difference analyses using ICV and total brain volume (not total cortical volume).

- The authors do not find a significant interaction between sex and total SA on S-A axis loadings. Given the lack of a significant interaction, I appreciate that the results section does not over-interpret the within-sex effects not being identical between males and females. However, the discussion section does somewhat over-interpret these findings in the paragraph starting on line 408; this should be remedied.

- Across the different analysis methods, the authors take the top 10% of connections etc. The manuscript would benefit from either a strong justification for choosing 10% (e.g., previous work has shown similar findings across multiple thresholds including 10%) and/or the inclusion of additional thresholds in their analyses.

I have the following more minor comments:

- The citations used when talking about previous resting-state studies focused on sex differences (e.g., line 90-92) are on the older side (e.g., Bluhm 2008, Biswal 2010). I would recommend including citations for more recent work as well, such as Ritchie 2018 in the UK Biobank.

- The manuscript states in multiple places that TBV does not include subcortical volumes. However, in the methods section subcortical structures are listed as having contribute to the TBV values. I assume the latter is a typo, but this should be remedied.

- The paragraph which starts on line 447 appears to be interpret the findings in relation to time-varying functional connectivity, which was not examined in this manuscript. I would recommend the authors make this more clear.

- The functional gradients and geodesic distances were calculated per hemisphere, where the former was calculated per hemisphere to allow for comparability with the latter. However, the MPC – which is also investigated in conjunction with functional gradients – was calculated across both hemispheres. There should be additional justification for this inconsistency and/or a brief test showing this inconsistency does not have any meaningful impact on the measures.

- Line 50-52: Citations are missing for this sentence.

- Line 433: I know the clause which begins 'which' is referring to association networks, but the structure of the sentence may make it seem like it's referring to sensory networks for some readers. I'd suggest rephrasing.

Additionally, I have the following suggestions to improve figure clarity:

- The circles used throughout to depict group differences would benefit from having the network labels depicted directly on the figure, instead of the reader needing to look back and forth between the color key and the circle. If possible, it would be best to only use the more standard acronyms (e.g., DMN, VA(N), DA(N)) and to fully spell out other network names (e.g., visual instead of 'V', limbic instead of 'L').

- Figure 3B should indicate somehow that none of the t-stats depicted are significant.

- The subfigures within Figure 4 do not appear to be labeled in the order they appear in the text, nor do they visually seem to 'hang together' the way the results in the text do. It would be beneficial to reorder the subfigures in Figure 4 to improve readability.

- Figure 4I would benefit from an indicator of which differences were significant.

- Figure 4J should list the network names on the x axis. It would be best to only use the more standard acronyms (e.g., DMN, VA(N), DA(N)) and to fully spell out other network names (e.g., visual instead of 'V', limbic instead of 'L').

Reviewer #3 (Remarks to the Author):

The goal of this study was two-fold: (1) examine the association of the S-A functional connectivity organization with cortical morphology and (2) examine sex differences in the S-A functional cortical organization across the sensorimotor axis and test whether these sex differences are correlated with sex differences in cortical morphology (i.e., total SA, MPC, geodesic distance) and in network topology (i.e., mean FC and within network dispersion). The authors find sex differences in the sensory-association (S-A) axis of functional cortical organization and largely based on their spatial correlation analyses, suggest that these are associated with sex differences in network topology rather than cortical morphometry. Although this is a very interesting study with beautiful figures, I believe that the authors can substantially improve the manuscript and the reach of the paper by addressing the following:

Major:

1. The authors are examining correlations and moderation effects. They should not be using causal language to describe their analyses, such as the verb "explains". This needs to be changed throughout the manuscript.

o Here is an example where you use the appropriate language in the discussion: "sex differences in the S-A axis were related to differences in FC profiles".

o In the intro you write "functional consequences" Why would global brain size be influencing functional connections and not vice versa? How can you assume this directionality?

2. Why were t statistics chosen over effect sizes? T statistics provide information on the significance of an association, but this will vary with several parameters (one vs two tail, df, etc).

This makes the t statistic difficult to interpret. Moreover, t statistics do not provide any information about the size of the difference or its practical significance. All t statistics should be updated to effect sizes, Cohen's d, or standardized betas, which can be meaningfully interpreted across your analyses and other studies.

3. Individuals with neurodevelopmental and psychiatric disorders are part of the typical population. You should estimate the S-A functional connectivity and sex differences in functional connectivity including these participants. If the results differ, you can remove them. Otherwise, these participants should be maintained in your study. Given that these analyses were not preregistered the selection of "sort of" healthy participants is questionable.

o If you're going to remove individuals with psychiatric disorders, you need to remove individuals with any condition or medication that influences brain chemistry...

o It seems strange to exclude subjects because of psychiatric disorders but not individuals who smoke or drink heavily, even without severe symptoms (whatever that means..., please explain).

o How many participants were excluded because of psychiatric or neurodevelopmental disorders?

o Have people previously reported that individuals with psychiatric disorders show differences in their S-A functional connections?

4. Figure 2E looks like "Simpson's Paradox" where a trend appears when groups are combined but disappears or reverses when trends are examined separately within each group. It does not seem meaningful to look at the trend across groups in the case of 2E. Please perform the analyses in each group separately and update your interpretation accordingly.

5. The introduction is long and hard to follow. The two main objectives are not clearly stated. The authors need to focus on presenting essential information related to their questions and improving the flow of their rationale.

o If differences in functional connectivity in sensory and association regions are the most robust across sex differences, why first focus on how sex differences in SA relate to morphometry rather than topology?

o Your argument for looking at cortical morphometry should be more explicit. The second paragraph is hard to digest, and the reader doesn't know where the author is going until the last phrase.

♣ You're not studying expansion here at all so I'm not sure why it's in your concluding sentence. "It is however unclear whether brain expansion..."

♣ How are the concluding sentences of paragraphs 2 and 3 different? What's the point of these paragraphs?

o In general, I suggest that the authors ensure that each paragraph is crucial for setting up the study.

o In the discussion, the paragraph on the rationale behind some analyses lines 364 -376 should be shortened and added to the introduction. The discussion is to discuss results not to explain the rationale of the analyses.

Minor:

1. "Sex differences in human brain size are robust and widely acknowledged [1-7]" You need to specify that differences in GLOBAL brain size are accepted because regional ones are not, as Eliot argues.

2. The authors should also be citing the paper from the original study instead of the commentary in response to Eliot's review by Williams et al., 2021. Particularly, when citing results from their study and not their commentary: "some sex differences still remain

statistically significant when the variance explained by total brain size is taken into account [7]”: Williams, C. M., Peyre, H., Toro, R., & Ramus, F. (2021). Neuroanatomical norms in the UK Biobank: The impact of allometric scaling, sex, and age. *Human Brain Mapping*, 42(14), 4623–4642. <https://doi.org/10.1002/hbm.25572>

3. The use of the word “fundamentally” across the manuscript is awkward. What are you trying to say?

4. You state on page 6 that “We began by computing the S-A axis as our measure of functional organization, given its relevance to cortical morphometry and sexual dimorphisms,” But before conducting your analyses you don’t know if the S-A will be relevant to study sex differences...

5. Line 386 – “minimize bias” in what?

6. What is the range of OR? it would be nice to get a sense of how big the differences you report are.

7. Given that the distribution of mean thicknesses across the cortex is associated with the S-A, it would have made sense to look at how mean cortical thickness relates to sex differences in S-A (Sydnor et al., 2021). It surprises me that none of the studies on S-A by Sydnor have been cited.

o Sydnor, V. J., Larsen, B., Bassett, D. S., Alexander-Bloch, A., Fair, D. A., Liston, C., ... & Satterthwaite, T. D. (2021). Neurodevelopment of the association cortices: Patterns, mechanisms, and implications for psychopathology. *Neuron*, 109(18), 2820-2846.

8. Dfispersion typo l.343

Reviewer #4 (Remarks to the Author):

This is a highly-derived analysis of male-female difference in functional connectivity that is attempting to rule out the influence of brain morphometry on such difference. The study utilizes the well-analyzed Human Connectome Project database (N=1000) to assess both sex difference in functional brain networks and the influence of structural sex difference (surface area, local myelin content, and geodesic distances) on such functional measures. The study finds first, that such structural measures substantially influence functional networks and second, paradoxically, that sex differences in functional connectivity are only modestly influenced by such structural factors.

Several features of the study are admirable, including the geodesic analysis, which appears to be a novel approach to control for sex differences in brain shape, and the inclusion of both thresholded and unthresholded findings, providing a more transparent view of the data. With that said, there are some weaknesses in the framing of the study, explanation of the primary dependent measure (the “sensory-association (S-A) axis), and assessment of effect sizes.

Global comments:

1. A major concern is the need for greater clarity about the primary measure used throughout the paper, the S-A axis, and why the authors hypothesize it would differ by sex/gender. Although referenced to Margulies et al. (2016), those earlier authors do not use this “S-A axis” terminology but refer to a “principal gradient” that extends between “primary and unimodal visual, somatosensory/motor, and auditory regions” on one end to the DMN on the other. The present study defines the “S-A axis” as extending “from visual to DMN regions” (line 596) so apparently is not including any other primary and unimodal sensory and motor areas. Thus,

“visual-DMN axis” would seem to be more precise terminology for this primary measure. Even then, it is not clear what is the significance of this axis. In the Introduction, this axis is described as “tethered” at the sensory end, but untethered at the association end, permitting a physical expansion of this axis as brain size increased in mammalian evolution. But why are the authors hypothesizing that this axis (or the degree of tethering?) differs between males and females? In other words, we need a clearer explanation of the significance of this axis and rationale for why it is being studied with respect to sex/gender difference. As it stands, this measure is very abstract and hard to relate to “greater female vulnerability to affective disorders” as is speculated in the concluding paragraph (line 504), or, for that matter, to any other behavioral/neuropsychiatric sex/gender difference.

I was hoping to gain a greater understanding from Figure 1, but part A (“mean S-A loadings”) just looks like a cortical map of primary, unimodal, and heteromodal association areas (albeit visual cortex is more “primary” than other sensory areas, per the chosen methodology). So what are we to make of the sex effect on this map (Figure 1B), where neither sex appears to be more loaded toward one or the other end of the gradient? There needs to be a common-language interpretation of Figure 1B, which is the crux of this paper. In my view, the more useful information is in Fig 1C which shows the comparative female vs. male loadings for each of the Yeo networks, but I'm not sure that is a novel analysis.

2. The paper is in need of an updated approach to male/female difference, which is never (especially in humans) a simple matter of genetic/hormonal “sex” versus experience/psychosocially-influenced “gender.” Biological organs, and especially the brain, are influenced by both sets of factors, especially by the time humans reach adulthood. Hence, the discussion of limitations on pp. 23-24 is overly simplistic. A sample as large as 1000 individuals will have both males and females of varying gender expression, whose life experiences will have shaped their functional brain networks. The authors acknowledge this entanglement by calling “unrealistic” the search for “clear-cut sexual dimorphism of brain structure and function.” Why, then, do they refer to “sexual dimorphism” in the Abstract and twice in the Introduction, a term that should be reserved for non-overlapping structural differences such as ovaries vs. testes? (See McCarthy et al., *J Neurosci*, 32:2241-2247, 2012.) A better way to acknowledge this entanglement is to use the conjoined term “sex/gender” or “gender/sex” to refer to the independent variable studied here and in any analysis of the human brain or behavior (Hyde et al., *Amer Psychol*, 2018, <http://dx.doi.org/10.1037/amp0000307>)

In this vein, are all participants in HCP genotyped to determine chromosomal sex, or are they divided according to “sex assigned at birth” as stated on line 471? Similarly, the statement that only gender involves “complex social and environmental influences” (line 473) is incorrect since every participant who grew up as a male or female experienced social and environmental influences associated with that assignment—experience that likely helped shape their functional brain networks. Similarly, when noting the environmental factors that can shape brain networks differently in males and females, stress (line 500) is but one of many. Others to mention include: caregiver strain, work strain, independence, risk-taking, emotional intelligence, social support and discrimination (Nielsen et al., *Biol Sex Diff*, 12:23, 2021). Any analysis of functional network differences between 29 y/o men and woman should therefore include descriptive statistics on their education, employment sector, income, caregiver status, at the least. I presume this data is available for the well-studied HCP cohort.

3. Please convert the t-scales to Cohen's d scales to give readers a better understanding of sex difference effect sizes. Since only 1.25% of all seed-based connections displayed a significant

sex difference (line 302) it appears these connectivity sex differences are overall quite modest, but it is hard to tell from the figures.

Specific comments:

Lines 23-24: change “anatomical dimorphism” to “size difference”

Lines 42 and 90: the three studies that are twice cited (Refs 5, 8, 9) actually report different findings with regard to sex differences in functional connectivity, so it is misleading to suggest that the literature leading up to this study is “consistent” – particularly since “sensory and association regions” basically covers the entire brain. (Ritchie et al. 2018 found higher connectivity in males in sensorimotor and visual networks, and in the DMN of females; Shanmugan et al. 2022 found sex difference greatest in ventral attention, DMN and frontoparietal networks; Weis et al. 2020 found areas most predictive of sex classification in rsfMRI were in the R. anterior and L. posterior cingulate and in the bilateral medial frontal and parietal areas). Hence, the results are not “robust” (line 88).

Lines 54-58: this scenario is unclear as written. If sensory tethering constrained cortical growth, why would “cortical expansion lead to the emergence of the S-A axis”?

Lines 72-75: The sentence beginning “Nevertheless...” should be cut: The issue of how to normalize brain size between individuals of different body size is unresolved and has a long and checkered history. (See the 1992 debate in Nature, vols 358, 359, and 360.) Both height and weight are poor metrics; some argue that fat-free body mass is the preferred measure (O’Brien et al., 2006; DOI: 10.1080/10673220600784119; Schoenemann, 2004; DOI: 10.1159/000073759) but this has not been used in large-scale brain imaging studies.

Regardless, this issue has no relevance to the existing study since the authors use only brain, not bodily morphometry as covariates in this study.

Line 81: Please indicate which cortical sex differences you are referring to in Ref. 6 and 16, since the present study is focused on the visual – association axis specifically.

Line 127: Cut “and sexual dimorphisms”

Line 159 (Figure 1): Some of the male/female loadings in part C do not seem to match with the maps in part B: e.g., the somatomotor and ventral attention network loadings on males.

Line 177: Which subcortical structures are excluded from TBV, as stated in this sentence? According to Methods, all of the basal ganglia, medial temporal lobe and thalamus are included.

Line 182: “For each measure of brain size...” But this analysis excludes geodesic, where the effect sizes (Fig 2I) are much larger than for Surface Area (Fig 2B).

Line 212 (Figure 2): Why is there no graph depicting the correlation between S-A and surface area, as there is for MPC and geodesic distance (2E and 2F)? Based on the t-values, it appears that geodesic distance is the superior morphometric determinant of S-A.

Line 239: Please plot the within-sex effects of Surface Area on S-A separately for males and females. If the SA sex difference does not account for the S-A sex difference, shouldn’t these be different slopes?

Line 264 (Figure 3): Why was only the sex*SA interaction tested (Fig 3B) and not sex*MPC or sex*mean geodesic distance interactions? Given the large influence of geodesic distance on S-A in Figure 1, it seems to be the most pertinent measure to assess in the context of sex difference. I appreciate that the t-map correlation (3F) is noisy and not significant, but the interaction term in the LMM may be more revealing.

Line 412: Please quantify the amount of variance in S-A sex difference that is accounted for by SA (rather than describing it as “some”). The same should be done for geodesic difference.

Line 441: Define “top” in this context (“top connections”)

Line 462: Is there any evidence that individual differences in S-A relate to cognition? If not, cut the clause “which may underpin cognitive differences.”

Lines 467-470: Sexual dimorphism is indeed an “unrealistic assumption” regarding brain structure and function, so why use the term in this paper? In addition, there are many gradations of gender beyond cis-trans that are unanalyzed in your data, so don’t single out trans people as complicating the analysis.

Lines 501-506: This seems like a big stretch, from sex differences in S-A axis to psychiatric gender disparities. Please explain how this would work. And if you are going to discuss female psychiatric vulnerabilities, you should also list male vulnerabilities that could just as easily relate to sex differences in functional brain networks (e.g., autism, ADHD, conduct disorder, substance use disorder).

Line 649: states they “excluded volumes of subcortical structures” in estimating TBV, but above this includes the volumes of the thalamus, caudate, putamen, pallidum, hippocampus, amygdala, and lateral ventricles. From the sounds of it, they only excluded the hypothalamus, brainstem and cerebellum. Please clarify.

Supplemental Figure S2: need numeric labels on the scale bar for both unthresholded and thresholded columns.

RESPONSE TO REVIEWERS (NCOMMS-23-59796)

We would like to thank the Editors and Reviewers for their positive evaluations, constructive comments, and for the opportunity to submit a revised manuscript. We believe that the resulting changes have significantly improved the clarity and quality of our work. We have addressed all questions and suggestions in a point-by-point fashion below and we have edited our manuscript and supplementary material accordingly (changes are highlighted in yellow).

Reviewer #1 (Remarks to the Author):

In this manuscript, Serio and colleagues investigated whether sex differences in cortical morphometry may explain sex differences in functional organization in the brain. Their analyses reveal negative findings, and instead suggest that differences in the sensory-association functional gradient between males and females are related to sex differences in functional connectivity. Major strengths of the study include the use of a large dataset and the consideration of multiple morphometric properties in the analyses. The manuscript reads well and the reported results are likely to be of broad interest to the neuroscientific community. Please find below my comments/questions on the work.

We thank the Reviewer for the appreciation of our work and the insightful comments, which we have addressed below.

1. Why was a 90% threshold used (i.e., only top 10% of functional connections considered) for the analyses? Do these results hold under other thresholds? Demonstrating that these results hold under multiple threshold (or at least yield the same (or similar) gradients) would strengthen the results.

Thank you for your question and suggestion. We used a 90% threshold (i.e., only considering the top 10% of functional connections) as this is the threshold that is typically used in the gradient literature to yield the sensory-association (S-A) axis (see Margulies et al. (2016) *PNAS*, the seminal paper presenting this method, but also many others including, for example, Hong et al. (2019) *Nature Communications* and Valk et al. (2020) *Science Advances*). Therefore, and as explained in our methods, the parameters we used to construct the S-A axis were selected for consistency and thus comparability with previous studies. Furthermore, the reproducibility of the 90% threshold has been assessed and approved in a study investigating the test-retest reliability of functional gradients (Knodt et al. (2023) *Human Brain Mapping*), as well as in a study comparing different thresholds, which found that more conservative thresholds resulted in higher reliability (Hong et al. (2020) *Neuroimage*).

Nevertheless, in order to address your comment and further test the robustness of our results as suggested, we conducted a supplementary sensitivity analysis to show that different thresholds (from 10% to 90%, in steps of 10%) all yield a similar S-A axis. We find that the mean S-A axis computed at the 90% threshold shows high correlations with mean S-A axes computed at other thresholds (with r values ranging from 0.84 to 0.93). We have now added these results to our manuscript, namely in Supplementary Figure 5 and in the methods section:

To confirm the robustness of the S-A axis computed at the 90% threshold, we further show with a sensitivity analysis that the mean S-A axis computed at the 90% threshold shows high correlations with mean S-A axes computed at different thresholds (from 10% to 90%, in steps of 10%), with r values ranging from 0.84-0.93 (see Supplementary Methods, Fig. 5).

Our motivations for using the 90% threshold are now also made clearer in the Results section:

We used this 90% threshold for consistency with previous studies [10, 35, 36] and for its high test-retest reliability and reproducibility [37, 38].

We refrained from comparing sex differences on the S-A axis computed at different thresholds given that we find in our main results that there are indeed sex differences in which connections represent the top 10% functional connections (i.e., the functional connections maintained with the 90% threshold), as reported in Figure 4F-G. Therefore, we expect that there would also be different sex differences in the “top connections” corresponding to different thresholds –given that we would be considering and comparing a set of new and different connections– which however goes beyond the aims of our study.

2. The use of the acronyms S-A and SA is confusing, please consider using just one of the two.

Thank you for flagging this unclarity – we now refer to total surface area without acronym.

3. Figure S2 is missing a label for the colorbar.

Thank you for noticing this omission – we added a β as label for the colorbar indicating standardized beta coefficients.

4. Table S1 - I believe there may be a typo in the total SA values listed (mean for males is shown as 1947.35 and mean for females is shown as 17150.67).

Thank you for spotting this typo on the female value – we corrected it to the following: 1715.07.

Reviewer #2 (Remarks to the Author):

In this paper, the authors thoroughly investigate the possible structural and functional underpinnings of sex differences in the sensory-association axis, which can be considered one of the major organizational properties of the functional connectome in humans. They find that the examined structural properties are associated with the S-A axis but do not appear to underpin sex differences in the S-A axis, despite exhibiting sex differences themselves. They additionally find sex differences in other functional connectivity properties, which may relate to sex differences in the S-A axis. The manuscript is clearly written, and the analyses are nicely rigorous. As a whole, I consider this manuscript to be an important contribution to the field that bolsters our understanding of the basic underlying organizational properties of the human brain.

We thank the Reviewer for the appreciation of our work and the helpful comments and suggestions, which we have addressed below.

My main comments are as follows:

1. The primary finding of the manuscript – that sex differences in S-A are not related to morphometric properties – is informative but also essentially a null result. It would therefore be helpful if the authors reframed that finding as what effect size they had power to detect. This would be similar to when genetics papers find no significant associations between their trait of interest and SNPs; instead of saying there is fundamentally association, they say that – if there is an association – it is no larger than (e.g., medium or small) effect size.

Thank you for expressing your concerns about power – we are happy to further clarify. We have not originally conducted a power analysis to inform our sample size given that the HCP sample of $N = 1000$ conforms to suggestions of sample sizes and power required in brain-wide association studies (see Marek et al. (2022) *Nature*). In fact, we did find effects of both sex and morphometric measures on the S-A axis, as well as sex effects on morphometric measures, suggesting that power was appropriate, at least for detecting sex effects.

The null results reported in our study refer to a lack of overlap between sex differences in S-A axis loadings and sex differences in morphometric measures, as well as the lack of sex by morphometric measure interaction effects on the S-A axis loadings. We believe the first null results to be particularly robust, in light of known inflations of association estimates due spatial correlations across the cortex (Alexander-Bloch et al. (2018) *Neuroimage*), and thus interpret it as a theoretically meaningful lack of association at the group level. To address your comment and compute power for the sex by morphometric measure interaction effects, we conducted a simulations-based post-hoc power analysis using SIMR (an R package for power analysis of generalized linear mixed models by simulation; Green & MacLeod (2015) *Methods in Ecology and Evolution*). Specifically, we used 200 simulations to compute power for the sex by morphometric measurement interaction effect on the S-A axis (corresponding to analyses shown in Figure 3B, G, and H), with different specifications of effect sizes (i.e., using 200 simulations per tested effect). Simulations show that we had 71.5% power to detect a $\beta = 0.2$ (small effect) and 95.5% power to detect a $\beta = 0.3$ (medium effect), suggesting that we were powered to detect medium sex by morphometric measure interaction effects. We also note, that specifying $\beta = 0.315$ in our simulations (i.e., the largest observed effect that we obtained in modelling the specified interaction terms) indicates 97% power.

That being said, we are cautious about formally including (post-hoc) power analyses in our manuscript as there is general concern over the conceptual meaningfulness of post-hoc power analyses (Quach et al. (2022) *General Psychiatry*).

2. One of the main findings is that sex differences in the S-A axis are associated with sex differences in functional connectivity profiles. For this analysis the authors assessed sex differences in pairwise functional connectivity (in the top 10% of connections) and interpret these sex differences as partially underpinning sex differences in S-A loadings. However, given that the S-A loadings derive from pairwise functional connectivity (in the top 10% of connections), the manuscript would benefit from a clearer description of how these findings are not at least partially circular.

Thank you for suggesting that this aspect of our work could be further clarified. There is indeed an element of circularity in our analyses, which we however consider as meaningful to our question. To further clarify this: In our study, we aimed to investigate whether sex differences in the S-A axis are associated with sex differences in measures of cortical morphometry. When we found that this is not the case, we dug deeper into the functional connectivity data itself in order to investigate where the sex differences in the S-A axis may come from at a functional (instead of morphometric) level. In other words, we specifically investigated which feature(s) of the functional connectivity data have sex differences that are associated with the sex differences that we identify in the S-A axis – or, put differently, which features of the functional connectivity data does our data reduction algorithm detect sex differences in, which are then reflected as sex differences in our low dimensional representation of the functional connectivity data.

We tested three different features here: mean functional connectivity strength, functional connectivity profiles (i.e., which pairwise functional connections are the top 10% connections of per seed area), and network topology. When probing which areas represent the top 10% functional connections –and whether there are sex differences in this feature of the functional connectivity data– we needed to consider the data that our data reduction algorithm had been applied on in order to investigate what may have influenced it (it would have been irrelevant to consider connections below the top 10% of functional connections here, as that would be data that our data reduction algorithm had not seen).

Investigating sex differences in connectivity profiles (i.e., in the strongest connections per seed region) at the individual-level is also insightful in and of itself, given that studies that threshold their functional connectivity data rarely further look into *which* connections are conserved at the individual level following thresholding –whether it being more traditional edge-wise thresholding of functional connectivity matrices or thresholding in the context of data reduction, as we have done (Knodt et al. (2023) *Human Brain Mapping*). Indeed, the comparison of thresholded functional connectivity strength at the individual or group level may not reflect a direct or meaningful comparison if the thresholded pairwise connections being compared across individuals/groups are not “fixed” connections between the same set of areas.

In our study, we find sex differences in the top 10% functional connections, which is a relevant observation because sex differences in the S-A axis could have, for example, reflected differences in functional connectivity strength instead, which is what is typically reported in the literature (Allen et al. (2011) *Frontiers in Systems Neuroscience*; Biswal et al. (2010) *PNAS*; Bluhm et al. (2008) *Neuroreport*; Scheinost et al. (2015) *Human Brain Mapping*; Weis et al. (2020) *Cerebral Cortex*; Ritchie et al. (2018) *Cerebral Cortex*; Shanmugan et al. (2022) *PNAS*). In fact, we found that sex differences in functional connectivity strength (of the top

10% connections) were *not* statistically associated with sex differences in the S-A axis, despite there being an element of circularity here as well (i.e., always looking at the same data – the top 10% functional connections). This observation further supports the idea that different features of the functional connectivity data may have variable relevance for explaining sex differences observed on a low dimensional representation of functional connectivity data (i.e., the S-A axis). The lack of association between sex differences in functional connectivity strength and sex differences in the S-A axis also strengthens our finding that sex differences in the top 10% connections may support sex differences observed on the S-A axis.

We appreciate that this was not made clear enough in our manuscript and have added clarifications in our Results section:

Concretely, we investigated which sex differences in features of FC may be detected by data reduction and reflected in the sex differences that we observe in the S-A axis loadings.

[...]

In this way, we specifically considered the data –at the individual level– that our data reduction algorithm has been applied on in order to probe whether the top 10% connections made by females and males may have differentially influenced the computation of the S-A axis and consequently sex differences in its loadings.

As well as in our Discussion section:

A such, our findings highlight the variable relevance of different features of FC data in representing sex differences in a low dimensional representation of functional organization, specifically suggesting that sex differences in the S-A axis may be better represented by FC profiles than FC strength.

3. The term ‘brain size’ is used in an unclear and potentially misleading manner in some sections of the manuscript. Specifically, I would argue that most readers would assume ‘brain size’ is referring to ICV or TBV if no additional specification is provided. However, in the abstract and elsewhere ‘brain size’ is used without additional specification to refer to total SA. Similarly, ‘TBV’ is used to refer to total cortical volume, not total brain volume; this could also be considered misleading. I understand why the authors chose to examine total SA and total cortical volume, but the terminology used throughout the abstract and manuscript should reflect what was examined. If the authors wish to discuss ‘brain size’ in the abstract without additional specification, then they should complete supplementary sex difference analyses using ICV and total brain volume (not total cortical volume).

Thank you for pointing out that the terminology we use may be misleading. We have clarified this by replacing our use of the term “total brain volume (TBV)” with “total cortical volume (TCV)” as suggested. We are now also more specific in using the terms “total surface area” and “different measures of brain size” instead of generally using the term “brain size” when referring to our analyses and results. We kept instances of the general term “brain size” when appropriate, e.g., in the introduction when referring to general sex differences in brain size, which motivate our study.

4. The authors do not find a significant interaction between sex and total SA on S-A axis loadings. Given the lack of a significant interaction, I appreciate that the results section does not over-interpret the within-sex effects not being identical between males and females. However, the discussion section does somewhat over-interpret these findings in the paragraph starting on line 408; this should be remedied.

Thank you for pointing out that our discussion appears to over-interpret our findings, as this was not our intention. We reformulated this paragraph in our discussion in order to avoid any hint of over-interpretation:

After establishing morphometric correlates of the S-A axis, we addressed our primary aim of probing the extent to which sex differences in functional cortical organization may be reflected by sex differences in cortical morphometry. Our findings overall suggest that morphometric differences between the sexes are altogether not substantial contributors of sex differences in the S-A axis of functional organization. We did not find any statistical spatial associations between patterns of sex differences in the S-A axis and patterns of sex differences in the MPC axis nor in the mean geodesic distance of connectivity profiles. Although we observed slightly diverging results when including –as opposed to excluding– total surface area as a covariate in our model testing for sex differences in S-A axis loadings, we did not find a statistically significant interaction between sex and surface area in reflecting S-A axis loadings. We also find no statistically significant sex by MPC axis interactions, but find few statistically significant sex by mean geodesic distance interactions. This can be understood as partly mirroring the sex differences in functional connectivity profiles that we further explain below, given that geodesic distances were averaged based on the connectivity profiles –more specifically the top 10% functional connections– which we find sex differences in. Altogether, the negligible relevance of cortical morphometry to sex differences in the S-A axis is striking given that morphometric properties appear per se to be associated with the S-A axis and to differ between sexes. The mechanisms underpinning different patterns of morphometric and functional sex differences may thus be independent from one another, suggesting that sex differences in functional cortical organization may extend beyond the connectome’s supporting shape and structure.

5. Across the different analysis methods, the authors take the top 10% of connections etc. The manuscript would benefit from either a strong justification for choosing 10% (e.g., previous work has shown similar findings across multiple thresholds including 10%) and/or the inclusion of additional thresholds in their analyses.

Thank you for addressing this point – see our answer and the actions taken under Reviewer #1 (point 1) on p.1-2 of this document.

I have the following more minor comments:

6. The citations used when talking about previous resting-state studies focused on sex differences (e.g., line 90-92) are on the older side (e.g., Bluhm 2008, Biswal 2010). I would recommend including citations for more recent work as well, such as Ritchie 2018 in the UK Biobank.

Thank you for suggesting more recent work. We have now added more recent work (i.e., Ritchie et al. (2018) *Cerebral Cortex*; Zhang et al. (2020) *Frontiers in Human Neuroscience*) to statements where we describe stronger FC in females within the DMN and in males within sensorimotor areas.

7. The manuscript states in multiple places that TBV does not include subcortical volumes. However, in the methods section subcortical structures are listed as having contribute to the TBV values. I assume the latter is a typo, but this should be remedied. Apologies for the confusion. In our measure of TBV, we initially wanted to include volumes that are “anatomically located within the cortical sheath” as originally explained in our methods (lines 647-650 of the original submission). However, we appreciate the contradiction that this entails with our desire to exclude subcortical volumes, and the confusion that it creates in our manuscript.

As stated in our answer to your earlier comment (Reviewer #2 (point 3) on p.5 of this document), we have changed the name of the TBV variable to total cortical volume (TCV) and removed all subcortical structures from this measure accordingly, which now only includes volumes of the *TotCort_GM_Vol* and *Tot_WM_Vol* FreeSurfer output measures. We have

corrected this in our Methods section, and have reconducted our analyses with the new TCV measure instead of TBV and updated all findings accordingly (specifically in Supplementary Figure 2 and Supplementary Table 1).

8. The paragraph which starts on line 447 appears to be interpret the findings in relation to time-varying functional connectivity, which was not examined in this manuscript. I would recommend the authors make this more clear.

Thank you for pointing out that the way we report our network dispersion results can be misleading. The measures we use to analyze group-level differences in network topography (i.e., within- and between-network dispersion) are descriptive measures of the level of integration/segregation within and between functional networks, i.e., depicting how the functional networks are spread along the S-A axis (see Bethlehem et al. (2020) *Neuroimage*). This is not a method that characterizes time-varying changes in network topology, but rather a method that helps us characterize group differences in network topology.

We understand that our original sentence describing our result “greater male dispersion (i.e., decreased similarity on the S-A axis) within the DMN” may seem like we are referring to a time-varying effect, which we have now edited to avoid this confusion:

We in fact observed greater male dispersion relative to females within the DMN. This finding suggests that areas belonging to the DMN are represented further apart on the S-A axis (i.e., showing less similarity in their FC profiles) in males relative to females, which is also consistent with previous findings of generally more segregated male networks [52].

In order to avoid further confusion when contextualizing the possible meaning of network topology at the cognitive and physiological levels, we specified that our findings pertain to group differences and group averages:

Concretely, network topology, which represents the organization of functional communities within and between functional networks [42], may reflect brain states [53]– and in our case, possible differences thereof at the group-level.

[...]

Our findings of sex differences in network topology may therefore pertain to intricate sex differences not only in group-averaged brain states at rest, which may underpin cognitive differences, but also in global energy expenditure, which would reflect physiological differences.

9. The functional gradients and geodesic distances were calculated per hemisphere, where the former was calculated per hemisphere to allow for comparability with the latter. However, the MPC – which is also investigated in conjunction with functional gradients – was calculated across both hemispheres. There should be additional justification for this inconsistency and/or a brief test showing this inconsistency does not have any meaningful impact on the measures.

Thank you for flagging this inconsistency. We followed your suggestion and, for more consistency, we now report the MPC axis measure computed per hemisphere as our main analysis, as well as reporting that computing the MPC axis per hemisphere versus across hemispheres does not have a meaningful impact on the results. We specified this in our methods section:

For consistency and comparability with the computation of FC gradients and mean geodesic distance of connectivity profiles, we also computed MPC gradients independently per hemisphere and subsequently concatenated the gradient loadings resulting from each hemisphere. We verified and confirmed the

stability MPC gradients when computing them per hemisphere versus at the whole brain level, as shown by the spatial correlation of mean gradient loadings ($r = 0.99$, $p_{spin} < .001$).

10. Line 50-52: Citations are missing for this sentence.

Thank you for flagging this. We used this first sentence to introduce the arguments that would be made later in the paragraph (which we give citations for) and we originally did not want to repeat the references for the same statements. We however understand that references are expected in that introductory sentence, and have therefore added them accordingly.

11. Line 433: I know the clause which begins ‘which’ is referring to association networks, but the structure of the sentence may make it seem like it’s referring to sensory networks for some readers. I’d suggest rephrasing.

We rephrased the sentence as suggested to make it clearer:

This is consistent with a previous study in youth reporting that these association networks show greater individual variability in their functional topography relative to lower-order sensory networks, whilst contributing the most to sex classification [8].

Additionally, I have the following suggestions to improve figure clarity:

12. The circles used throughout to depict group differences would benefit from having the network labels depicted directly on the figure, instead of the reader needing to look back and forth between the color key and the circle. If possible, it would be best to only use the more standard acronyms (e.g., DMN, VA(N), DA(N)) and to fully spell out other network names (e.g., visual instead of ‘V’, limbic instead of ‘L’).

Thank you for your suggestions to improve figure clarity. We have added the functional network labels directly on the donut plots as suggested. We have decided against spelling out the network names of the visual and limbic networks for consistency and space reasons, but ensured that the acronyms are clearly defined both in the color keys and in the figure legends. Here is an example of the updated donut plots and color key (Figure 1):

13. Figure 3B should indicate somehow that none of the t-stats depicted are significant.

We have added “unthresholded” to the panel label and it is also specified in the figure legend as such:

*There were no statistically significant sex*total surface area interaction effects after FDR correction.*

14. The subfigures within Figure 4 do not appear to be labeled in the order they appear in the text, nor do they visually seem to ‘hang together’ the way the results in the text do. It would be beneficial to reorder the subfigures in Figure 4 to improve readability.

We reordered the panels of Figure 4 and made sure that their order corresponds to the order in which they are mentioned in the text.

15. Figure 4I would benefit from an indicator of which differences were significant.

Figure 4I shows the results of the analysis testing pairwise sex differences in between-network dispersion, none of which were statistically significant – as indicated in the text:

LMMs did not show any statistically significant sex differences in between-network dispersion for any of the network pairs (Fig. 4I).

and in the figure legend:

I | β -values for the sex contrast in between-network (BN) dispersion for each pairwise Yeo network comparison, where blue represents higher male BN dispersion and red represents higher female BN dispersion (no statistically significant sex effects after spin permutation and Bonferroni correction; $p_{spin} < .001$)

We have included this figure in the main results despite its lack of statistically significant effects in order to transparently disclose effect sizes and directions.

16. Figure 4J should list the network names on the x axis. It would be best to only use the more standard acronyms (e.g., DMN, VA(N), DA(N)) and to fully spell out other network names (e.g., visual instead of ‘V’, limbic instead of ‘L’).

As done for our donut plots following your previous suggestion, we have added the network names directly on the x-axis of the figure.

For more clarity and based on external feedback we have received for this plot, we have also decided to display only half of the violins in order to more intuitively portray that we are showing distributions:

Reviewer #3 (Remarks to the Author):

The goal of this study was two-fold: (1) examine the association of the S-A functional connectivity organization with cortical morphology and (2) examine sex differences in the S-A functional cortical organization across the sensorimotor axis and test whether these sex differences are correlated with sex differences in cortical morphology (i.e., total SA, MPC, geodesic distance) and in network topology (i.e., mean FC and within network dispersion). The authors find sex differences in the sensory-association (S-A) axis of functional cortical organization and largely based on their spatial correlation analyses, suggest that these are associated with sex differences in network topology rather than cortical morphometry. Although this is a very interesting study with beautiful figures, I believe that the authors can substantially improve the manuscript and the reach of the paper by addressing the following:

We thank the Reviewer for their evaluation of our work and suggestions for improvement and reach of the paper, and hope to have addressed all concerns.

Major:

1. The authors are examining correlations and moderation effects. They should not be using causal language to describe their analyses, such as the verb “explains”. This needs to be changed throughout the manuscript.

Here is an example where you use the appropriate language in the discussion: “sex differences in the S-A axis were related to differences in FC profiles”.

Thank you for this comment. Indeed, we do not test for causality and understand that some of the terminology that we used can be misleading. Therefore, as suggested, we have replaced the use of the verb “explains” with terms such as “related to” and “associated with” when referring to associations/correlations in our aims and results throughout our manuscript.

In the intro you write “functional consequences” Why would global brain size be influencing functional connections and not vice versa? How can you assume this directionality?

With regards to brain size associations with functional connectivity, we believe that there may be some level of bidirectionality in this association and that it is therefore difficult to establish (uni)directionality. In the specific statement from our introduction that you are referring to, we were not referring to the evolutionary relationship between structure and function and the question of what comes first, but we were instead referring to the idea that structure physically supports function (along the lines of Pang et al. (2023) *Nature* – who argue for “Geometric constraints on human brain function”, specifically constraints imposed by brain shape, which we later cite in our introduction as a rationale for our analyses). In our introduction, we thus specifically referred to the robustness of sex differences in brain size and highlight that there is a lack of robustness in the “corresponding” sex differences in brain function, which would be expected given the premise that structure supports function, regardless of the directionality of this association. Nevertheless, this was only the angle we took to frame our motivations for this study, as the first sentence of our introduction. As you have correctly pointed out, we did not specifically test for causality in our work and therefore this question of directionality goes beyond the aims of our study.

We however acknowledge that our phrasing may put too much stress on the directionality of the association, and therefore changed it from “downstream functional consequences” to “functional implications”.

2. Why were t statistics chosen over effect sizes? T statistics provide information on the significance of an association, but this will vary with several parameters (one vs two tail, df , etc). This makes the t statistic difficult to interpret. Moreover, t statistics do not provide any information about the size of the difference or its practical significance. All t statistics should be updated to effect sizes, Cohen's d , or standardized betas, which can be meaningfully interpreted across your analyses and other studies.

Thank you for pointing this out. We agree that measures of effect sizes can be more meaningfully interpreted than t statistics and have therefore changed all of our t statistics to standardized beta coefficients accordingly.

3. Individuals with neurodevelopmental and psychiatric disorders are part of the typical population. You should estimate the S-A functional connectivity and sex differences in functional connectivity including these participants. If the results differ, you can remove them. Otherwise, these participants should be maintained in your study. Given that these analyses were not preregistered the selection of "sort of" healthy participants is questionable.

If you're going to remove individuals with psychiatric disorders, you need to remove individuals with any condition or medication that influences brain chemistry...

It seems strange to exclude subjects because of psychiatric disorders but not individuals who smoke or drink heavily, even without severe symptoms (whatever that means..., please explain).

How many participants were excluded because of psychiatric or neurodevelopmental disorders?

Have people previously reported that individuals with psychiatric disorders show differences in their S-A functional connections?

We apologize for the confusion: We did not screen out any individuals with neurodevelopmental and psychiatric disorders from the existing Human Connectome Project (HCP) sample. In our methods, we described the sampling procedure (including the inclusion/exclusion criteria) of the HCP, which is beyond our control given that we are using pre-collected data from an open dataset. To clarify this in our Methods section, we have now specified that subjects with neurodevelopmental and psychiatric disorders were excluded "*from the HCP study recruitment protocol*".

Nevertheless, we thank you for your pertinent point that individuals with neurodevelopmental and psychiatric disorders are part of the typical population. We have suggested the need for a more inclusive sample in our study limitations (in our Discussion section):

Finally, our findings are limited to sex differences in a healthy sample and would benefit from being replicated in a more inclusive sample that is more representative of the overall population. This would also be additionally informative considering the notable sex differences observed in populations that would typically be excluded from healthy samples, for example individuals with neurodevelopmental and psychiatric disorders [65]. Nevertheless, the structure-function associations that we investigated in this work are rather fundamental, and their essence should thus be fairly well captured in our large (healthy) sample.

4. Figure 2E looks like "Simpson's Paradox" where a trend appears when groups are combined but disappears or reverses when trends are examined separately within each group. It does not seem meaningful to look at the trend across groups in the case of 2E.

Please perform the analyses in each group separately and update your interpretation accordingly.

Thank you for your suggestion. We performed the correlation between the mean S-A axis loadings and the mean MPC axis loadings separately for each sex as requested, which yield the following results:

- Males: $r = 0.17$, $p < .001$, $p_{spin} = 0.070$
- Females: $r = 0.23$, $p < .001$, $p_{spin} = 0.018$
- (Across groups: $r = 0.20$, $p < .001$, $p_{spin} = .037$)

These findings show that when looking at trends separately within groups, the correspondence between the S-A and MPC axes in males does not pass the statistical significance threshold after spin permutation. However, we would not over-interpret this null finding for multiple reasons. First, our analyses show that there does not seem to be any statistically significant sex*MPC interaction in predicting S-A axis loadings in any of the 400 parcels (as shown in Figure 3G), suggesting that the relationship between the S-A and MPC axes does not differ between the sexes in a statistically significant manner. Second, the correlations between the S-A and MPC axes within sex that we performed above suggest similar effects that go in the same direction (i.e., not seemingly supporting the case of Simpson's Paradox), which are both statistically significant before the spin test controlling for spatial autocorrelation, and that only do not pass the statistical significance threshold in the male group when conducting the spin test, possibly because the male sample is slightly less powered compared to the female sample (i.e., $n_{males} = 464$; $n_{females} = 536$) in addition to the effect size being slightly lower in males. Furthermore, given the smoothness of cortex-wide patterns, we do expect some level of spatial autocorrelation to still meaningfully reflect similarities in patterns exhibited by two brain maps. Therefore, we believe that our findings hold as originally presented in our manuscript.

5. The introduction is long and hard to follow. The two main objectives are not clearly stated. The authors need to focus on presenting essential information related to their questions and improving the flow of their rationale.

Thank you for your suggestions to improve the clarity and flow of our introduction. We have addressed your suggestions point-by-point and have made changes to our manuscript accordingly. Please see more detailed answers below.

If differences in functional connectivity in sensory and association regions are the most robust across sex differences, why first focus on how sex differences in SA relate to morphometry rather than topology?

Thank you for your question. To clarify, our main aim was to test whether sex differences in functional cortical organization are associated with sex differences in cortical morphometry. Thus, our focus was not on localizing sex differences and describing how they relate to network topology, but rather on establishing associations between functional cortical organization and cortical morphometry (and specifically, associations between their sex differences). This is also the order in which we conducted our analyses and presented our results (i.e., starting by constructing the S-A axis and testing for sex differences in the S-A axis (Figure 1), then establishing associations between the S-A axis and morphometric features (Figure 2), followed by testing for associations between sex differences in the S-A axis and sex differences in morphometric features (Figure 3)). In fact, only when we found that sex differences in the S-A axis were not associated with sex differences in cortical morphometry, did we test for other possible explanations of what the sex differences in the S-A axis may reflect (here considering sex differences in intrinsic functional connectivity, more specifically in functional connectivity strength, connectivity profiles, and network topology (Figure 4)). Therefore, we report our

analyses and findings in the order that they were conducted, following a specific rationale, and our introduction aims to transparently set up this rationale accordingly.

Your argument for looking at cortical morphometry should be more explicit. The second paragraph is hard to digest, and the reader doesn't know where the author is going until the last phrase.

- **You're not studying expansion here at all so I'm not sure why it's in your concluding sentence. "It is however unclear whether brain expansion..."**
- **How are the concluding sentences of paragraphs 2 and 3 different? What's the point of these paragraphs?**

In general, I suggest that the authors ensure that each paragraph is crucial for setting up the study.

Thank you for pointing out paragraphs that could be clearer in setting up the background and rationale for our study. In paragraph 2, we aimed to explain the relevance of brain size for cortical organization, using –as supporting evidence– patterns of scaling via expansion across evolution and development. We have now made our argument more explicit at the beginning of this paragraph:

In order to understand how sex differences in brain size may pertain to sex differences in brain function, it is first necessary to understand the relevance of brain size for overall functional cortical organization.

Paragraph 3 moves on to describe patterns of sex differences in cortical morphometry, which reflect scaling patterns along the S-A axis, as described in paragraph 2.

In the concluding sentence of paragraph 2, we referred to brain expansion in order to introduce the possible consequences of this phenomenon (i.e., in this way introducing paragraph 3). We however understand that it may be misleading given that our study does not focus on expansion but rather on brain size per se, and we have therefore reformulated this sentence for the focus to be on size/scaling (see below).

Indeed, the concluding sentences of paragraphs 2 and 3 were very similar. To improve the flow of our introduction and make the relevance of each paragraph clearer, we rephrased and broken down the last sentences of paragraph 2 and 3 as such:

Paragraph 2 (relevance of brain size for cortical organization):

Through the increase of overall brain size, the differential expansion of sensory and association areas could thus be an important product of mammalian evolution and development, further reflecting the hierarchical functional differentiation of these regions. In fact, this differential scaling and reorganization of regions along the S-A axis also appears to reflect patterns of sex differences in cortical morphometry, that is, cortical shape and size.

Paragraph 3 (sex differences in cortical morphometry):

Yet, how exactly sex-specific differences in cortical morphometry may be relevant to differences in intrinsic brain function has not been directly explored.

In the discussion, the paragraph on the rationale behind some analyses lines 364 - 376 should be shortened and added to the introduction. The discussion is to discuss results not to explain the rationale of the analyses.

Thank you for pointing this out. As suggested, we have moved the sentences explaining that different measures of brain size yield different magnitudes of sex differences to the introduction:

Different measures of brain size are commonly used in the literature (such as intracranial volume, total brain volume, and total surface area). Although these measures highly covary and are often used interchangeably, they quantify different morphometric features of the brain, with sex differences in “brain size” ranging from 8% to 13% depending on the selected measure [6]. The size and direction of sex effects also vary by neuroanatomical property, such as different tissue types, brain regions, and features (including cortical thickness, gyrification, and surface area) [16].

We have also restructured the second paragraph of our discussion accordingly:

Considering the common use of different measures of brain size in the literature, which yield different magnitudes of sex differences depending on the selected measure [6], we tested the effects of different measures of brain size on the S-A axis, namely ICV, TCV, and total surface area. Here, given that total surface area had the most widespread effects on functional organization, we deemed it the most appropriate measure of brain size for our study and further included it as a covariate in our models throughout our analyses. The relevance of total surface area for our study is also supported by the theoretical assumptions motivating our research question, namely the relevance of cortical shape and geometry in constraining brain wide functional dynamics [28-30] and thus sex differences in these features potentially underpinning sex differences in the S-A axis. In showing the diverging statistical effects of different measures of brain size, our findings highlight the risk of introducing noise when including an inadequate measure of brain size, particularly when statistically controlling for brain size in the detection of sex effects on brain structure and function [31, 43-46]. The complex heterogeneity of neuroanatomical properties constituting brain size should not be undermined, also considering that morphometric features vary differently as a function of age, whereby for example total brain volume but not ICV is affected by atrophy [6]. As such, future research on sex differences should also carefully select the measure of brain size that is most conceptually and empirically pertinent to the research question under study in order to avoid introducing noise in the analyses.

Minor:

6. “Sex differences in human brain size are robust and widely acknowledged [1-7]” You need to specify that differences in GLOBAL brain size are accepted because regional ones are not, as Eliot argues.

Thank you for this specification – we added the word “*global*” as suggested.

7. The authors should also be citing the paper from the original study instead of the commentary in response to Eliot’s review by Williams et al., 2021. Particularly, when citing results from their study and not their commentary: “some sex differences still remain statistically significant when the variance explained by total brain size is taken into account [7]”: Williams, C. M., Peyre, H., Toro, R., & Ramus, F. (2021). Neuroanatomical norms in the UK Biobank: The impact of allometric scaling, sex, and age. *Human Brain Mapping*, 42(14), 4623–4642. <https://doi.org/10.1002/hbm.25572>

Thank you for spotting this – we have changed the citation in our manuscript accordingly.

8. The use of the word “fundamentally” across the manuscript is awkward. What are you trying to say?

Thank you for pointing this out – the use of this word was indeed inadequate. We wanted to underline the lack of systematic relationship between sex differences in functional organization and sex differences in cortical morphometry, and therefore we have replaced the word “fundamentally” with “systematically”.

9. You state on page 6 that “We began by computing the S-A axis as our measure of

functional organization, given its relevance to cortical morphometry and sexual dimorphisms,” But before conducting your analyses you don’t know if the S-A will be relevant to study sex differences...

Thank you for spotting this lack of clarity. In that sentence, we intended to underline the relevance of sensory and association areas for sex differences, which are at the extremities of the S-A axis (rather than suggesting that the S-A axis is itself relevant to sex differences). We have now edited this sentence in our manuscript to clarify this:

We began by computing the S-A axis as our measure of functional organization, given its relevance to cortical morphometry and given that sex differences in functional connectivity are typically found in regions situated at the extremities of this axis, i.e., in sensory and association regions.

10. Line 386 – “minimize bias” in what?

Bias is indeed the wrong word here, thank you for pointing it out. What we were trying to express is the fact that choosing the correct measure of brain size is essential to avoid the introduction of noise in the analyses. For example, ICV remains largely stable across the lifespan after reaching its peak as opposed to total brain volume. Another example is that of total surface area, which decreases at a different (slower) rate than mean cortical thickness after both reach their peaks (Bethlehem et al. (2022) *Nature*). As such, it is important to use the measure of the “brain size” that is the most pertinent to the question/mechanism under study in order to avoid capturing variance in the wrong brain structure, i.e., to avoid introducing noise.

As such, we have edited replaced “*minimize bias*” with “*avoid introducing noise in the analyses*”.

11. What is the range of OR? it would be nice to get a sense of how big the differences you report are.

Thank you for asking to specify this. Given that the OR should be interpreted differently when $OR < 1$ or $OR > 1$ (where $OR > 1$ indicates a given region’s greater male odds and $OR < 1$ indicates a given region’s greater female odds), please find below the range of the OR for statistically significant sex differences in both these instances:

- Range for $OR < 1$: 0.00 - 0.64
- Range for $OR > 1$: 1.56 - 25.36

We have added this detail to our results section:

For connections showing statistically significant sex differences, we found an OR ranging from 0.00 - 0.64 in the case of greater female odds ($OR < 1$), and an OR ranging from 1.56 to 25.36 in the case of greater male odds ($OR > 1$).

12. Given that the distribution of mean thicknesses across the cortex is associated with the S-A, it would have made sense to look at how mean cortical thickness relates to sex differences in S-A (Sydnor et al., 2021). It surprises me that none of the studies on S-A by Sydnor have been cited.

Sydnor, V. J., Larsen, B., Bassett, D. S., Alexander-Bloch, A., Fair, D. A., Liston, C., ... & Satterthwaite, T. D. (2021). Neurodevelopment of the association cortices: Patterns, mechanisms, and implications for psychopathology. *Neuron*, 109(18), 2820-2846.

Thank you for your suggestion to include mean cortical thickness in our analyses. Indeed, cortical thickness shows similar spatial patterns to patterns of the S-A axis, as do many other brain features (e.g., evolutionary hierarchy, aerobic glycolysis, cerebral blood flow, gene

expression, cognitive function, externopyramidalization) (Sydnor et al. (2021) *Neuron*). We did not include mean cortical thickness in our analyses and how it relates to sex differences in the S-A axis because cortical thickness does not conceptually pertain to our rationale and narrative. In fact, in our introduction, we describe an evolutionary account of the importance of cortical expansion (and thus brain size) for the functional organization of the human cortex, and propose mean geodesic distance and microstructural organization as additionally relevant morphometric features of the cortex to consider following this rationale.

For the conceptual reasons mentioned above, we thus did not include cortical thickness in our manuscript. For your interest and as displayed below, we have applied part of our analysis pipeline to cortical thickness in order to test for associations between cortical thickness and the S-A axis (**C**; $r = 0.42$, $p_{\text{spin}} = 0.004$), as well as associations between sex differences in cortical thickness and sex differences in the S-A axis (**D**; $r = 0.17$, $p_{\text{spin}} = 0.093$). These results show similar patterns as those found in our manuscript, both in showing an association between a morphometric feature and the S-A axis, and in showing a lack of association between sex differences in that given morphometric feature and sex differences in the S-A axis.

With regards to your suggestion of citing work by Valery Sydnor, we have added the reference that you have shared (reference number 15) in our introduction:

Patterns of expansion across cortical regions along the S-A axis are also observed across human development, with a more markedly distributed areal expansion across frontoparietal association regions relative to limbic and sensorimotor areas [14] – see also [15] for a comprehensive review of the S-A axis' neurodevelopment.

13. Dfispersion typo l.343

Thank you for spotting this - we have corrected it.

Reviewer #4 (Remarks to the Author):

This is a highly-derived analysis of male-female difference in functional connectivity that is attempting to rule out the influence of brain morphometry on such difference. The study utilizes the well-analyzed Human Connectome Project database (N=1000) to assess both sex difference in functional brain networks and the influence of structural sex difference (surface area, local myelin content, and geodesic distances) on such functional measures. The study finds first, that such structural measures substantially influence functional networks and second, paradoxically, that sex differences in functional connectivity are only modestly influenced by such structural factors.

Several features of the study are admirable, including the geodesic analysis, which appears to be a novel approach to control for sex differences in brain shape, and the inclusion of both thresholded and unthresholded findings, providing a more transparent view of the data. With that said, there are some weaknesses in the framing of the study, explanation of the primary dependent measure (the “sensory-association (S-A) axis), and assessment of effect sizes.

We thank the Reviewer for their evaluation and constructive and helpful comments that have further strengthened our work.

Global comments:

1. A major concern is the need for greater clarity about the primary measure used throughout the paper, the S-A axis, and why the authors hypothesize it would differ by sex/gender. Although referenced to Margulies et al. (2016), those earlier authors do not use this “S-A axis” terminology but refer to a “principal gradient” that extends between “primary and unimodal visual, somatosensory/motor, and auditory regions” on one end to the DMN on the other. The present study defines the “S-A axis” as extending “from visual to DMN regions” (line 596) so apparently is not including any other primary and unimodal sensory and motor areas. Thus, “visual-DMN axis” would seem to be more precise terminology for this primary measure.

Thank you for your careful consideration of the primary measure used in our work. We are happy to further clarify our decision of calling our measure the S-A axis.

The seminal study by Margulies et al. (2016) *PNAS*, applying data dimensionality reduction to functional connectivity matrices in order to yield a low dimensional representation of functional organization, has led many studies to use this method, which have however used different terminology for the derived measure, such as “principal gradient” (Margulies et al. (2016) *PNAS*), “sensory-association axis” (Meng et al. (2022) *Communications Biology*; Tooley et al. (2022) *Journal of Neuroscience*), “sensory-to-association cortical axis” (Knodt et al. (2023) *Human Brain Mapping*), “sensorimotor-association cortical axis” (Sydnor et al. (2023) *Nature Neuroscience*), “sensory-to-transmodal hierarchy” (Huntenburg et al. (2021) *Neuroimage*), “sensory-fugal axis” (Lee et al. (2023) *Human Brain Mapping*), “unimodal-to-multimodal gradient” (Tong et al. (2022) *Nature Communications*). Despite the different terminology, these studies all refer to the same principle of brain organization, which is essentially an axis that depicts the hierarchical differentiation of cortical regions, with sensory and association regions at its poles. We believed that calling our measure the principal gradient, as done by Margulies et al. (2016), would not be helpful for readers who are not familiar with this method. We therefore chose the most descriptive and self-explanatory term that best reflects this general principle of functional organization (on which we also build the rationale for our study in our introduction) – hence the term “S-A axis”.

As you have correctly noticed, the measure yielded by our analyses indeed spans from visual to DMN regions, with somatosensory/motor regions loading more towards the center of the axis. It is not uncommon that the measure derived from reducing the dimensionality of functional connectivity matrices varies a bit based on the sample and parcellation scheme used (for example, we used Schaefer 400, whilst Margulies et al. (2016) conducted their analyses on native surface space (i.e., including 32492 vertices per hemisphere, that is 59412 vertices excluding the medial wall). Nevertheless, as shown by many publications using this dimensionality reduction method on functional connectivity data, as well as studies specifically assessing the reliability and replicability of this method (Knodt et al. (2023) *Human Brain Mapping*; Hong et al. (2020) *Neuroimage*), this method yields a robust and reproducible gradient that consistently spans from sensory to association regions. Despite “visual-DMN axis” being a more precise terminology as you suggest, we believe that it would cause confusion as it would add yet another term to the already existing multitude of terms used to depict the same phenomenon. As such, given that visual and DMN regions, the extremities of our gradient, mirror seminal findings whilst belonging to sensory and association regions respectively, we believe that calling our measure S-A axis is valid, increases clarity with regards to our motivations for conducting this study, and preserves homogeneity with existing literature.

2. Even then, it is not clear what is the significance of this axis. In the Introduction, this axis is described as “tethered” at the sensory end, but untethered at the association end, permitting a physical expansion of this axis as brain size increased in mammalian evolution. But why are the authors hypothesizing that this axis (or the degree of tethering?) differs between males and females? In other words, we need a clearer explanation of the significance of this axis and rationale for why it is being studied with respect to sex/gender difference. As it stands, this measure is very abstract and hard to relate to “greater female vulnerability to affective disorders” as is speculated in the concluding paragraph (line 504), or, for that matter, to any other behavioral/neuropsychiatric sex/gender difference.

Thank you for letting us know that the significance of the S-A axis should be further clarified in the context of our study on sex differences in functional organization. Given that our study builds on and integrates different strands of research (i.e., cortical morphometry, functional organization, and sex differences), we understand that we were not clear enough in our introduction in separating these concepts before suggesting how they may relate to one another, which is the rationale for our study.

Your comment referring to the tethering hypothesis pertains to paragraph 2, which aims to introduce a major principle of functional cortical organization (i.e., the hierarchical differentiation between sensory and association regions) in an evolutionary context whilst showing the relevance of brain size for functional organization (which is important for us to explain, given that there are robust sex differences in brain size). Here, we describe the tethering hypothesis to give a theoretical evolutionary account of the emergence of the S-A axis (reflecting this major principle of functional organization) across millions of years of evolution. Therefore, the tethering hypothesis should not be interpreted as the basis for sex differences, but as the description of a principle of functional organization which has developed throughout evolution, and which is present in all humans (regardless of their sex) as well as in other species such as macaques (Margulies et al. (2016) *PNAS*; Valk et al. (2022) *Nature Communications*) and marmosets (Tong et al. (2022) *Nature Communications*), and to some lesser extent mice (Huntenburg et al. (2021) *Neuroimage*). Given the confusion caused by this paragraph, we clarified its aims as well as how it pertains to our study of sex differences:

In order to understand how sex differences in brain size may pertain to sex differences in brain function, it is first necessary to understand the relevance of brain size for overall functional cortical organization.

[...]

Through the increase of overall brain size, the differential expansion of sensory and association areas could thus be an important product of mammalian evolution and development, further reflecting the hierarchical functional differentiation of these regions. In fact, this different scaling and reorganization of regions along the S-A axis also appears to reflect patterns of sex differences in cortical morphometry, that is, cortical shape and size.

After establishing the S-A axis as a valid measure describing a major principle of functional organization, interpreting sex differences on this axis is a separate matter. You have asked why we would hypothesize sex differences in the “degree of tethering” of sensory and association regions, referring to the evolutionary theory describing the emergence of the S-A axis. However, it is important to note that we do not expect differences in the S-A axis of different individuals (whatever their sex) to reflect a phenomenon that can be compared to the untethering of association areas from sensory areas that has unfolded throughout millions of years of evolution. Previous studies have shown that the S-A axis varies between individuals, relating to semantic cognition (Shao et al. (2022) *Cortex*), as well as varying in psychiatric conditions (i.e., in autism; Hong et al. (2019) *Nature Communications*), and has further been associated with ageing and measures of general cognition (i.e., fluid and crystallized intelligence; Knodt et al. (2023) *Human Brain Mapping*). In fact, (sex) differences in functional connectivity will be reflected on the S-A axis, given that the S-A axis is simply a low dimensional representation of functional connectivity. Despite appearing abstract at first, Knodt et al. (2023; *Human Brain Mapping*) have called the S-A axis “biologically meaningful” and “more interpretable” than high dimensional connectivity matrices, in that the S-A axis captures a well-established principle of hierarchical cortical organization, and allows to take a more system-level perspective when assessing differences in S-A axis loadings. In our study, we use Figure 4 to further interpret the meaning of sex differences in the S-A axis, aiming to understand what features of intrinsic functional connectivity were represented in the S-A axis as sex differences. There, we found sex differences in the S-A axis to reflect sex differences in functional connectivity profiles and network topology, more specifically greater dispersion in males relative to females within the DMN. We have dedicated a full paragraph in our discussion to discuss the possible meaning of these specific findings:

We in fact observed greater male dispersion relative to females within the DMN. This finding suggests that areas belonging to the DMN are represented further apart on the S-A axis (i.e., showing less similarity in their FC profiles) in males relative to females, which is also consistent with previous findings of generally more segregated male networks [52]. These network-specific topological sex differences may be related to greater female odds of connections within the DMN, and greater male odds of somatomotor connections with other networks. Concretely, network topology, which represents the organization of functional communities within and between functional networks [42], may reflect brain states [53] – in our case, possible differences thereof at the group-level. Network topology has also been associated with different cognitive features including arousal [54], awareness and consciousness [55], behavior and task performance [56], and cognitive flexibility [57]. The balance between integration and segregation is complex, dynamic, and necessary to maintain the brain’s metastability [58] by reaching a point of equilibrium between global organization and local specialization [48]. The brain is a highly interconnected and metabolically expensive organ, and its organization is required to dynamically balance topological efficiency and energy utilization in response to transient cognitive and physiological demands [59]. Our findings of sex differences in network topology may therefore pertain to intricate sex differences not only in group-averaged brain states at rest, which may underpin cognitive differences, but also in global energy expenditure, which would reflect physiological differences.

With regards to our reference to “greater female vulnerability to affective disorders” in the last sentence of our discussion, this is indeed not something that we have assessed in our study, but rather general and meaningful context to frame the relevance of understanding sex differences in functional brain organization. In our study, we found that sex differences in functional organization do not appear to be associated with sex differences in brain shape or size, and thus more work needs to be done to assess other potential drivers of sex differences in functional organization (for example neuroendocrine and environmental factors, as mentioned), which may underpin the differences in functional competitiveness profiles and network topology that we observe, and which may be relevant for gaining a better understanding of sex differences in psychiatric disorders in future work. We have edited the end of our discussion to make this clearer:

Ultimately, investigating the mechanisms underpinning individual differences in human brain structure and function is crucial to gain a deeper understanding of risk and resilience to psychiatric disorders. Studying sex differences in the brain is one important component of this, given the differential vulnerabilities to psychiatric disorders as a function of sex and gender across the lifespan [70, 71].

For your further interest, differences in the S-A axis have previously been associated with neurodiversity and psychopathology (i.e., autism; Hong et al. (2019) *Nature Communications*, also see review by Sydnor et al. (2021) *Neuron* on "Neurodevelopment of the association cortices: Patterns, mechanisms, and implications for psychopathology").

I was hoping to gain a greater understanding from Figure 1, but part A (“mean S-A loadings”) just looks like a cortical map of primary, unimodal, and heteromodal association areas (albeit visual cortex is more “primary” than other sensory areas, per the chosen methodology). So what are we to make of the sex effect on this map (Figure 1B), where neither sex appears to be more loaded toward one or the other end of the gradient? There needs to be a common-language interpretation of Figure 1B, which is the crux of this paper. In my view, the more useful information is in Fig 1C which shows the comparative female vs. male loadings for each of the Yeo networks, but I'm not sure that is a novel analysis.

We are happy to further clarify: Figure 1 is actually not the crux of this paper, but rather depicts initial analyses that we needed to perform in order to test our hypothesis that sex differences in functional cortical organization may be related to sex differences in cortical morphometry. Therefore, in Figure 1A, we show the computed (mean) S-A axis, and in Figure 1B and C, we show the resulting sex differences in the S-A axis (note that Figure 1C is a breakdown of the sex differences shown in Figure 1B, in order to more explicitly show the networks involved). Despite the results shown in Figures 1B and C actually being novel (given that, to our knowledge, no other study has tested for sex differences in a low dimensional representation of functional connectivity), the aim of our study was not to “locate” sex differences on the S-A axis, but rather to see whether those sex differences may be (spatially) related to sex differences in cortical morphometry. We acknowledge that the relevance of Figure 1 was not made clear enough based on your comments and we have thus added clarification in our Results section:

In order to investigate whether sex differences in functional organization may be related to sex differences in cortical morphometry, we first needed to construct our measure of functional organization and test for related sex differences.

We appreciate that it is not intuitive to interpret the meaning of sex differences in S-A axis loadings. Despite the “location” of sex differences on the S-A axis per se not being the aim of

our study, we do interpret sex differences in S-A axis loadings when considering sex differences in the dispersion of functional networks (which are computed from the loadings of cortical areas, i.e., based on the relative “positions” of cortical areas on the S-A axis), as this is a more interpretable way of understanding differences on the S-A axis. As mentioned above, we do this in Figure 4 and further interpret these findings at a conceptual level in our discussion.

3. The paper is in need of an updated approach to male/female difference, which is never (especially in humans) a simple matter of genetic/hormonal “sex” versus experience/psychosocially-influenced “gender.” Biological organs, and especially the brain, are influenced by both sets of factors, especially by the time humans reach adulthood. Hence, the discussion of limitations on pp. 23-24 is overly simplistic. A sample as large as 1000 individuals will have both males and females of varying gender expression, whose life experiences will have shaped their functional brain networks. The authors acknowledge this entanglement by calling “unrealistic” the search for “clear-cut sexual dimorphism of brain structure and function.” Why, then, do they refer to “sexual dimorphism” in the Abstract and twice in the Introduction, a term that should be reserved for non-overlapping structural differences such as ovaries vs. testes? (See McCarthy et al., *J Neurosci*, 32:2241-2247, 2012.) A better way to acknowledge this entanglement is to use the conjoined term “sex/gender” or “gender/sex” to refer to the independent variable studied here and in any analysis of the human brain or behavior (Hyde et al., *Amer Psychol*, 2018, <http://dx.doi.org/10.1037/amp0000307>) In this vein, are all participants in HCP genotyped to determine chromosomal sex, or are they divided according to “sex assigned at birth” as stated on line 471? Similarly, the statement that only gender involves “complex social and environmental influences” (line 473) is incorrect since every participant who grew up as a male or female experienced social and environmental influences associated with that assignment—experience that likely helped shape their functional brain networks. Similarly, when noting the environmental factors that can shape brain networks differently in males and females, stress (line 500) is but one of many. Others to mention include: caregiver strain, work strain, independence, risk-taking, emotional intelligence, social support and discrimination (Nielsen et al., *Biol Sex Diff*, 12:23, 2021). Any analysis of functional network differences between 29 y/o men and woman should therefore include descriptive statistics on their education, employment sector, income, caregiver status, at the least. I presume this data is available for the well-studied HCP cohort.

Thank you for your considerations as well as for sharing publications on the sex/gender issue. We fully agree that both genetic/hormonal “sex” and experience/psychosocially-influenced “gender” may affect biological organs, especially the brain, and we are sorry to read that this was not made clear enough in our limitations, given that our strong belief in the influence of both factors is precisely why we included this limitation in the first place. We acknowledge that gender-related factors may also have influenced brain-related measures in our sample, and have clarified this in our limitations (also citing the paper that you have shared; i.e., Hyde et al. (2018) *American Psychologist* (reference number 61)):

As such, we capitalized on a binary variable to detect group-level effects, being conscious however that complex social and environmental experiences that go beyond a binary categorization are also likely to have influenced the brains of individuals in our sample throughout their lifetime [61], to an extent that we could however not quantify in this work.

The sex variable was self-reported, and we did not verify it with genetic information. We have added this information to our Methods section, in accordance with a passage in Dhamala et al.

(2022) *Human Brain Mapping* (reference number 73), which also investigated sex differences in the HCP sample and report the following:

Despite the use of the term gender in the HCP Data Dictionary, we use the term sex in this article given that the HCP study collected self-reported information on biological sex instead of gender identification, as reported elsewhere [73]. We have not used genetic information to verify the self-reported biological sex.

We have also briefly specified this at the end of our Introduction, when introducing the HCP sample.

To this end, we used multimodal imaging data (including resting state functional MRI and structural T1 and T2 images) of the Human Connectome Project (HCP) S1200 release [32], consisting of healthy young adults self-reporting their biological sex.

After thoughtful consideration, we have decided against using the term “sex/gender” or “gender/sex” in our study because we find that it may be misleading in making it seem as if we have considered gender-related variables in our study, which we have not, and prefer to transparently comment on this shortcoming in our limitations instead (although we understand that these terms refer to the entanglement of sex/gender constructs). In fact, the aims of our study were to investigate the correspondence (i.e., shared variance) between sex differences in cortical morphometry and sex differences in cortical functional organization regardless of variance explained by gender, which we further clarified in our limitations:

Nevertheless, we intentionally focused on the dichotomous variable of sex given that our study aimed to investigate the correspondence (i.e., shared variance) between sex differences in cortical morphometry and sex differences in cortical functional organization, regardless of variance explained by gender.

For the reasons above, we believe that that the term sex/gender would create confusion in the specific context of our study.

We agree that “clear-cut sexual dimorphism in brain structure and function” are an unrealistic assumption, as stated in our manuscript. Our use of the word “dimorphism” in our abstract and introduction was intended to be descriptive of specific findings of robust sex differences in brain morphometry observed when dividing the population in two distinct categories (male and female), which is how research has historically and predominately been treating this variable. Nevertheless, we appreciate that continuing to use this term contradicts our own statement and beliefs, possibly even harming the field, which is why we have replaced the term “dimorphism” with “difference”.

Thank you for listing a range of environmental factors can shape brain networks differently across the sexes. We chose to mention stress, and underlined that it is only an example, given its known epigenetic mechanisms and relevance for psychopathology, which we discuss in the following sentence:

Environmental factors should equally be considered, for example stress, which has also been found to contribute to sex differences in brain function and psychopathology via epigenetic mechanisms [68, 69]. Ultimately, investigating the mechanisms underpinning individual differences in human brain structure and function is crucial to gain a deeper understanding of risk and resilience to psychiatric disorders. Studying sex differences in the brain is one important component of this, given the differential vulnerabilities to psychiatric disorders as a function of sex and gender across the lifespan [70, 71].

Thank you for suggesting to include detailed descriptive statistics on our sample. We have

created an additional Supplementary Table (Supplementary Table 3) for the data that was available in the HCP sample, which we also now refer to in our methods:

The sample included 284 monozygotic twins (MZ), 184 dizygotic twins (DZ), 443 non-twin siblings, and 89 unrelated individuals, and the sociodemographic breakdown of the participants is additionally reported in Supplementary Table 3.

4. Please convert the t-scales to Cohen’s d scales to give readers a better understanding of sex difference effect sizes. Since only 1.25% of all seed-based connections displayed a significant sex difference (line 302) it appears these connectivity sex differences are overall quite modest, but it is hard to tell from the figures.

Thank you for this suggestion. In accordance with another comment made by Reviewer #3 (point 2) (see p.11 of this document), we have changed all of our *t* statistics to standardized beta coefficients for better interpretations of effect sizes. As also mentioned on p.15 of this document in response to Reviewer #3 (point 11), we have also added the range of ORs for the analysis of sex differences in functional connectivity profiles.

We would argue that sex differences in the brain are generally modest, and sometimes within-group variance is even larger than between-group mean effects (Wierenga et al. (2019) *Journal of Cognitive Neuroscience*), as stated in our introduction. Nevertheless, modest effect sizes do not suggest that sex differences are not meaningful. Given the substantial sex differences observed in the clinical setting (e.g., Rubinow & Schmidt (2019) *Neuropsychopharmacology*), we believe that it is necessary to consider and follow-up on these modest effects.

Specific comments:

5. Lines 23-24: change “anatomical dimorphism” to “size difference”

As mentioned above, we have replaced mentions of “dimorphisms” in our paper with “difference”.

6. Lines 42 and 90: the three studies that are twice cited (Refs 5, 8, 9) actually report different findings with regard to sex differences in functional connectivity, so it is misleading to suggest that the literature leading up to this study is “consistent” – particularly since “sensory and association regions” basically covers the entire brain. (Ritchie et al. 2018 found higher connectivity in males in sensorimotor and visual networks, and in the DMN of females; Shanmugan et al. 2022 found sex difference greatest in ventral attention, DMN and frontoparietal networks; Weis et al. 2020 found areas most predictive of sex classification in rsfMRI were in the R. anterior and L. posterior cingulate and in the bilateral medial frontal and parietal areas). Hence, the results are not “robust” (line 88).

Thank you for noticing this. We originally used the word robust to indicate that males consistently show higher intrinsic functional connectivity in sensory regions and females in association regions, pointing to this general pattern and directionality findings. However, we appreciate that the specific sensory and association regions reported as showing sex differences in the cited studies differ. Therefore, to avoid confusion, we have removed the adjective “robust” as you suggested.

7. Lines 54-58: this scenario is unclear as written. If sensory tethering constrained cortical growth, why would “cortical expansion lead to the emergence of the S-A axis”?

Thank you for suggesting that this sentence could be further clarified. With the verb “constrain”, we do not mean that sensory systems completely inhibited cortical expansion but

rather acted as “anchors” or pulling forces, and thus shaped cortical organization through cortical expansion accordingly. We have now further clarified this:

According to the tethering hypothesis, the brain’s sensory systems, acting as anchors, may have exerted constraining pressures during the growth of the developing ancestral mammalian cortex [13].

8. Lines 72-75: The sentence beginning “Nevertheless...” should be cut: The issue of how to normalize brain size between individuals of different body size is unresolved and has a long and checkered history. (See the 1992 debate in Nature, vols 358, 359, and 360.) Both height and weight are poor metrics; some argue that fat-free body mass is the preferred measure (O’Brien et al., 2006; DOI: 10.1080/10673220600784119; Schoenemann, 2004; DOI: 10.1159/000073759) but this has not been used in large-scale brain imaging studies. Regardless, this issue has no relevance to the existing study since the authors use only brain, not bodily morphometry as covariates in this study.

Thank you for this insight, we have removed this sentence accordingly.

9. Line 81: Please indicate which cortical sex differences you are referring to in Ref. 6 and 16, since the present study is focused on the visual – association axis specifically.

Thank you for asking. We have specified the regions showing sex differences in cortical morphometry from those references:

In fact, sex differences in cortical morphometry are partly located at the anchors of the S-A axis: A meta-analysis identified volumetric sex differences in multiple cortical regions, including the anterior and posterior cingulate gyri, precuneus, right frontal pole, inferior and middle frontal gyri, insular cortex, Heschl’s gyrus, and lateral occipital cortex [6]. Another study found greater grey matter volume in females in prefrontal and superior parietal cortices, whilst males showed greater volumes in ventral occipitotemporal regions [18]. These findings further depict the apparent relevance of the S-A axis as an axis of morphometric variability between the sexes.

We would like to further clarify that the aim of that paragraph in the introduction is to stress the global patterns of sex differences in morphometry and functional connectivity that are female-biased in association regions and male-biased in sensory regions. As such, we do not believe that a one-to-one correspondence between the areas mentioned above and our axis going from visual to association regions is necessary. Instead, we aim to show that these findings support sex differences in the hierarchical differentiation between sensory and association regions.

10. Line 127: Cut “and sexual dimorphisms”

As mentioned above, we have rephrased any mention of “sexual dimorphisms”.

11. Line 159 (Figure 1): Some of the male/female loadings in part C do not seem to match with the maps in part B: e.g., the somatomotor and ventral attention network loadings on males.

Thank you for expressing your concerns. Please find below the breakdown of the number of cortical areas showing sex differences in S-A axis loadings (by functional network and sex) displayed in Figure 1C:

- Visual - Male: 1, Female: 6
- Somatomotor - Male: 7, Female: 4
- Default mode network - Male: 12, Female: 16
- Dorsal attention - Male: 2, Female: 2
- Ventral attention - Male: 17, Female: 3

- Limbic - Male: 0, Female: 5
- Fronto parietal - Male: 16, Female: 2

For reference, we have illustrated statistically significant sex differences in areas belonging to somatomotor and ventral attention networks when isolating these networks (including a reminder of the original Figure 1B and C, as well as the Yeo 7 network legend key):

12. Line 177: Which subcortical structures are excluded from TBV, as stated in this sentence? According to Methods, all of the basal ganglia, medial temporal lobe and thalamus are included.

We apologize for the confusion here. As mentioned in response to Reviewer #2 (point 3) on p.5 of this document, as well as in response to Reviewer #2 (point 7) on p.6-7 of this document, we have changed the name of our total brain volume (TBV) variable to total cortical volume (TCV), and removed all subcortical structures from this measure, now only including volumes of the *TotCort_GM_Vol* and *Tot_WM_Vol* FreeSurfer output measures (and corrected this in our methods section). We have also reconducted our analyses with the TCV measure instead of TBV and updated the findings accordingly in Supplementary Figure 2 and Supplementary Table 1.

13. Line 182: “For each measure of brain size...” But this analysis excludes geodesic, where the effect sizes (Fig 2I) are much larger than for Surface Area (Fig 2B).

Thank you for your comment. Mean geodesic distance is not a measure of brain size, but a measure of the mean distances of the cortical profiles (i.e., the distances between areas representing top 10% connections per cortical region). The three different measures of brain size that we tested are intracranial volume (ICV), total cortical volume (TCV) and total surface area, as mentioned at the beginning of that paragraph. As you have correctly pointed out, geodesic distance is tested later (and displayed in the same figure, i.e., Figure 2F, I, J) as a morphometric measure.

14. Line 212 (Figure 2): Why is there no graph depicting the correlation between S-A and surface area, as there is for MPC and geodesic distance (2E and 2F)? Based on the t-values, it appears that geodesic distance is the superior morphometric determinant of S-A.

Thank you for your comment, we are happy to further clarify. Indeed, we do not show a correlation between the S-A axis and total surface area because it is not possible given the dimensions of the data arrays that we are working with. For the S-A axis, the MPC axis and mean geodesic distance, we have arrays that have 400 values (i.e., one value per cortical area). For total surface area, we only have one value, that is the total surface area of the brain, and therefore we could correlate one value to 400 values.

15. Line 239: Please plot the within-sex effects of Surface Area on S-A separately for males and females. If the SA sex difference does not account for the S-A sex difference, shouldn't these be different slopes?

Thank you for your comment. The plotted within-sex effects of surface area on the S-A axis are available in supplementary materials (Supplementary Figure 3D-F), as mentioned in text on p.12). We do not expect these within-sex effects of surface area on the S-A axis to have different slopes between males and females given that different slopes would suggest a statistically significant sex*total surface area interaction, which we do not find (as explained in text on p.12 and shown in Figure 3B).

16. Line 264 (Figure 3): Why was only the sex*SA interaction tested (Fig 3B) and not sex*MPC or sex*mean geodesic distance interactions? Given the large influence of geodesic distance on S-A in Figure 1, it seems to be the most pertinent measure to assess in the context of sex difference. I appreciate that the t-map correlation (3F) is noisy and not significant, but the interaction term in the LMM may be more revealing.

Thank you for your question. We originally did not test for the sex*MPC axis and sex*geodesic distance interaction effects given that we wished to test for spatial associations between the sex differences found in the S-A axis and the sex differences found in the morphometric features (i.e., MPC axis and mean geodesic distance). The reason why we tested for a sex by total surface area interaction is that testing for spatial associations between sex differences in total surface area and sex differences in the S-A axis is not possible (total surface area only has one value, as explained in response to your previous comment (point 14)). In fact, the spatial association between sex differences in the S-A axis and sex differences in morphometric features does not test exactly the same relationship as the interaction between sex and morphometric features (i.e., the moderation (by sex) of the effect of a morphometric feature on the S-A axis). We test the latter for total surface area for lack of a better option.

We appreciate that –despite the reasons mentioned above– showing the results of sex*MPC axis and sex*geodesic distance interaction effects may be informative. For this reason, we have now computed these effects. We find no statistically significant sex*MPC interactions and find 35 cortical regions showing statistically significant sex*geodesic distance interactions, in bilateral medial prefrontal cortex, temporal regions, and left dorsolateral regions (outlined in black on Figure 3H). This can be understood as partly mirroring the sex differences in functional connectivity profiles shown in Figure 4F-H, as there is some level of circularity here. Indeed, given that there are sex differences in the top 10% functional connections –for example, with females showing more top 10% connections involving the DMN and males showing more top 10% connections involving somatomotor regions, which are respectively known to make long- and short-range connections (Wang et al. (2023) *Cerebral Cortex*), it is to be expected

that there are sex differences in mean geodesic distance and some level of sex by mean geodesic distance interaction effect on the S-A axis loadings. Considering that the spatial associations between sex differences in the S-A axis and sex differences in morphometric measures (i.e., the MPC axis and mean geodesic distance) are null, as well as the null moderation of sex in the effect of total surface area on the S-A axis, we believe that the overall finding of our work remains the same, namely that sex differences in morphometry do not systematically reflect sex differences in the S-A axis, and that sex differences in connectivity profiles are much more indicative of sex differences in the S-A axis.

We have now added the sex*MPC axis and sex*geodesic distance interaction effects to Figure 3 (panels G and H respectively):

We refer to these new findings in our results section:

Despite our aims of testing for spatial associations between sex differences in cortical morphometry and sex differences in the S-A axis, we additionally modeled interaction terms of sex by MPC axis loadings (Fig. 3G) and sex by mean geodesic distance (Fig. 3H) within the original LMMs to test these interaction effects on S-A axis loadings. Here, we found no statistically significant sex by MPC axis interaction effects, and find 35 cortical regions showing statistically significant sex by geodesic distance interaction effects, specifically in the bilateral medial prefrontal cortex, temporal regions, and left dorsolateral regions (delineated in black on Fig. 3H).

as well as in our methods:

We also modelled two additional interaction terms within the original LMMs of sex by MPC axis loadings and sex by mean geodesic distance to show their effects on the S-A axis loadings.

We have added a discussion of the meaning of these effects for our study in the Discussion section, as described above, and believe that these added findings further underline the importance of sex differences in connectivity profiles:

We also find no statistically significant sex by MPC axis interactions but find few statistically significant sex by mean geodesic distance interactions. This can be understood as partly mirroring the sex differences in functional connectivity profiles that we further explain below, given that geodesic distances were averaged based on the connectivity profiles –more specifically the top 10% functional connections– which we find sex differences in. Altogether, the negligible relevance of cortical morphometry to sex differences in the S-A axis is striking given that morphometric properties appear per se to be associated with the S-A axis and to differ between sexes.

[...]

The strongest functional connections (top 10% connections) of females seemed to involve the DMN more than in males, whereas males displayed more top connections involving the somatomotor networks relative to females. This is also consistent with the few regional statistically significant sex by mean geodesic distance interaction effects on S-A axis loadings. Indeed, given that sensory and association regions are respectively known to make short- and long-range connections [25], it is expectable to observe sex differences in mean geodesic distance and some level of a sex by mean geodesic distance interaction effects on the S-A axis loadings given the observed sex differences in FC profiles involving these specific regions.

17. Line 412: Please quantify the amount of variance in S-A sex difference that is accounted for by SA (rather than describing it as “some”). The same should be done for geodesic difference.

Thank you for your comment. In general, we have now changed our reporting of effects from the linear mixed effects models from t -values to standardized beta coefficients as previously mentioned, which makes the assessment of effect sizes more interpretable overall in our Results section.

In the specific case that you mention in your comment, we are referring to a change in the r coefficient for the correlations between β maps reflecting sex differences yielded by linear mixed effects models that include different covariates (i.e., case #1: controlling for all morphometric variables (total surface area, MPC axis, and mean geodesic distance) vs controlling only for total surface area ($r = 0.95$, $p_{\text{spin}} < .001$, Supplementary Figure 4A); case #2: controlling for all morphometric variables (total surface area, MPC axis, and mean geodesic distance) vs not controlling for any morphometric variable ($r = 0.81$, $p_{\text{spin}} < .001$, Supplementary Figure 4B). Given that the correlation of β maps changes from $r = 0.95$ to $r = 0.81$ when excluding total surface area from one of the models, we simply wanted to transparently disclose that some minor amount of variance seems to be explained by total surface area (although in this specific instance, we cannot quantify this amount of variance). We do however clearly report in Figure 3B that the sex*total surface area interaction has an effect size up to $\beta = 0.232$ across cortical areas, despite none of the cortical areas showing statistically significant interaction effects of sex*total surface area, hence our statement that the amount of variance explained by total surface area in sex effects is minor.

In our discussion, we avoid including numerical statistical results, given that standard practice is to include numerical results in the results section only. Nevertheless, we have changed the sentence that you referred to in your comment to make it clearer that, despite observing slightly diverging results when including –as opposed to excluding– total surface area as a covariate in our model testing for sex differences in S-A axis loadings, we did not find a statistically significant interaction between sex and surface area in reflecting S-A axis loadings, which is the overall important finding that we are communicating:

Although we observed slightly diverging results when including –as opposed to excluding– total surface area as a covariate in our model testing for sex differences in S-A axis loadings, we did not find a statistically significant interaction between sex and total surface area in reflecting S-A axis loadings.

18. Line 441: Define “top” in this context (“top connections”)

We have clarified this (i.e., top 10%) and generally rephrased that sentence:

The strongest functional connections (top 10% connections) of females seemed to involve the DMN more than in males, whereas males displayed more top connections involving the somatomotor networks relative to females

19. Line 462: Is there any evidence that individual differences in S-A relate to cognition? If not, cut the clause “which may underpin cognitive differences.”

Thank you for your question. As previously mentioned on p.18-20 of this document in response to your point (Reviewer #4, point 2), previous studies have shown that the S-A axis varies between individuals, relating this variability to semantic cognition (Shao et al. (2022) *Cortex*). The S-A axis has further been associated with age and measures of general cognition (i.e., fluid and crystallized intelligence; Knodt et al. (2023) *Human Brain Mapping*).

20. Lines 467-470: Sexual dimorphism is indeed an “unrealistic assumption” regarding brain structure and function, so why use the term in this paper? In addition, there are many gradations of gender beyond cis-trans that are unanalyzed in your data, so don’t single out trans people as complicating the analysis.

As mentioned above, we have replaced mentions of “dimorphisms” in our paper with “differences”.

Thank you for your expressing concerns and we sorry to read that our limitation can be interpreted as singling out transgender individuals as complicating the analysis, as it was not our intention. Despite the different gradations of gender beyond “cis-trans”, we referred to transgender individuals in our limitations given the limited existing literature on brain size differences in relation to gender (we had thus cited the paper Wiersch et al. (2023). Accurate sex prediction of cisgender and transgender individuals without brain size bias. *Scientific Reports*, whose findings were specifically aligned with the research question of our study). However, we understand that this can be misunderstood and we have therefore edited the sentence that you are referring to in your comment. We have also addressed the issue of complex social and environmental experiences that go binary categorization to our limitation, as mentioned before, also citing the paper that you have shared in your comment #3 (Hyde et al. (2018) *American Psychologist* (reference number 61). Please find below our revised discussion of the sex/gender limitation, incorporating all the aforementioned changes:

Firstly, by focusing on biological sex, we neglected possible effects of gender on functional organization and its morphometric correlates, as there are multiple factors contributing to sex differences, including both biological and social factors. Findings may indeed appear more nuanced if we moved beyond the unrealistic assumption of a clear-cut sexual dimorphism of brain structure and function [60], further considering the relevance of gender, steroid hormones, and the role of X and Y chromosomes [61]. Nevertheless, we intentionally focused on the dichotomous variable of sex given that our study aimed to investigate the correspondence (i.e., shared variance) between sex differences in cortical morphometry and sex differences in cortical functional organization, regardless of variance explained by gender. As such, we capitalized on a binary variable to detect group-level effects, being conscious however that complex social and environmental experiences that go beyond a binary categorization are also likely to have influenced the brains of individuals in our sample throughout their lifetime [61], to an extent that we could however not quantify in this work.

21. Lines 501-506: This seems like a big stretch, from sex differences in S-A axis to psychiatric gender disparities. Please explain how this would work. And if you are going to discuss female psychiatric vulnerabilities, you should also list male vulnerabilities that could just as easily relate to sex differences in functional brain networks (e.g., autism, ADHD, conduct disorder, substance use disorder).

Thank you for your comment. In the last sentence of our discussion that you are referring to, we did not mean to single out the S-A axis, but rather we wished to generally underline the importance of investigating sex differences in functional organization, as this may ultimately help us better understand sex differences in the predisposition to psychiatric disorders. We thus mentioned female psychiatric vulnerabilities to affective disorders as a specific example. However, we understand that this is one-sided and somewhat specific, and we have therefore edited the end of our discussion to make our statement more general:

Ultimately, investigating the mechanisms underpinning individual differences in human brain structure and function is crucial to gain a deeper understanding of risk and resilience to psychiatric disorders. Studying sex differences in the brain is one important component of this, given the differential vulnerabilities to psychiatric disorders as a function of sex and gender across the lifespan [70-71].

22. Line 649: states they “excluded volumes of subcortical structures” in estimating TBV, but above this includes the volumes of the thalamus, caudate, putamen, pallidum, hippocampus, amygdala, and lateral ventricles. From the sounds of it, they only excluded the hypothalamus, brainstem and cerebellum. Please clarify.

We again apologize for the confusion here. Please see our reply to your comment (Reviewer #4, point 12) on p.25.

23. Supplemental Figure S2: need numeric labels on the scale bar for both unthresholded and thresholded columns.

Thank you for pointing out this omission. We have added the labels on the scale bars of Supplementary Figure 2, which pertains to both the unthresholded and the thresholded columns.

Reviewer #1 (Remarks to the Author):

I thank the authors for revising their manuscript. They have now addressed all of my initial concerns and their additional analyses provide further evidence to support their conclusions. I am happy to recommend this manuscript for publication.

I do have one additional minor suggestion for the authors. I recommend authors consider using the term assigned sex (or sex assigned at birth) instead of biological sex. I understand prior work with the HCP dataset (and other work in neuroimaging) has used 'biological sex', but recent developments suggest the use of 'assigned sex' and 'sex assigned at birth' is more appropriate (<https://www.ncbi.nlm.nih.gov/pmc/articles/PMC10523819/>).

Reviewer #2 (Remarks to the Author):

I appreciate the authors' thoughtful and thorough response to my previous comments.

I have one last minor comment, based on my initial comment #1 ("The primary finding of the manuscript – that sex differences in S-A are not related to morphometric properties – is informative but also essentially a null result. ..."). I still think including in the manuscript the minimum detectable effect size would be informative, similar to initial GWAS that did as such and shaped our understanding of the expected effect sizes of disease-associated SNPs. However, I acknowledge this may be a field difference and I appreciate that the authors included these effect size calculations in their reviewer response. Particularly given that the reviewer response will be publicly posted online anyway (given Nature Communications' peer review process), I am okay with these calculations not being included in the manuscript.

As a whole, the authors have satisfactorily addressed my comments.

Reviewer #3 (Remarks to the Author):

the authors have appropriately addressed my comments and suggestions.

Reviewer #4 (Remarks to the Author):

I am still struggling with the primary aim of this study: although it makes sense to test for a relationship between morphometry and the S-A axis, the reasons for expecting a sex difference in the S-A axis are not well-supported (see below). The last part of the Results, wherein sex differences in functional connectivity profiles are analyzed, is not novel, nor did this analysis actually test whether such FC differences are the basis of sex differences in S-A axis, although it seems obvious they are related given that they are derived using similar functional correlation methods.

This revision did not answer my overall concern about the significance of sex difference in the S-A axis, which is such a broad way of looking at the cerebral cortex that it is challenging to relate

to any of the well-studied behavioral and clinical group-level differences between women and men. By definition, the S-A axis includes all of the cerebral cortex except for motor areas. Although expansion of this axis may be important in evolution and development, the authors do not make a convincing case for why they expect it to differ between men and woman. And while they spend much of the Introduction (lines 68-99) arguing that sex differences in regional cortical volumes are consistent with sex difference in the overall S-A axis, these regional differences have proven markedly discrepant across recent studies (see Line 93 comment below). So this is not a strong rationale for their experiment, and its actual result finding (Fig. 3) is consequently less interesting.

Another concern is that this study seems to over-hype what is actually a very modest sex difference. On Line 349 it is noted that only 1.25% of the 160,000 functional connections showed a significant sex difference. So it seems a stretch to call this a “divergent system-level organization of functional networks” between males and females [Line 573]. It’s also a far extrapolation to suggest that the findings in this paper will contribute meaningfully to understanding sex and gender differences in psychiatric vulnerability [lines 581-583].

As requested in my prior review, it’s important to state the many dimensions in which environmental factors might drive male and female functional networks to differ, beyond just “stress” [line 577]. I recommended then and reiterate my request that the authors report (in the Supplement) the male and female averages for any of the following factors which are moderated by gender and could be influencing the results: 1) income, 2) years of education, 3) physical activity level, and 4) caregiver/parental status, or any other variables that HCP collects which may differentially affect brain health or function in 29 year-old men vs. women.

Finally, I found the writing to be somewhat convoluted, more so than the prior draft, leading to a lack of clarity in the aims and significance of the findings. I’ve indicated a few examples below to illustrate this.

Specific comments:

Line 23: It’s hard to see brain size as “robustly” different between the sexes, given that the distributions of TBV overlap by nearly 50% between females and males (per Ref 5).

Line 93: According to a more recent paper by Eliot (<https://doi.org/10.1186/s13293-024-00585-4>) the regional brain volume sex differences reported by Williams et al. [Ref. 7] and other studies (including Refs. 6 and 18) have not been well-replicated, including the anterior and posterior cingulate, precuneus, inferior and middle frontal gyri, insula and lateral occipital cortex. But such claims are relied upon to rationalize this study, which states on Line 93 that such “findings further depict the apparent relevance of the S-A axis as an axis of morphometric variability between the sexes.”

Line 449: Example of convoluted writing: “Indeed, given that sensory and association regions are respectively known to make short- and long-range connections [25], it is expectable to observe sex differences in mean geodesic distances and some level of a sex by mean geodesic distance interaction effects on the S-A axis loadings given the observed sex differences in FC profiles involving these specific regions. A such, our findings highlight the variable relevance of different features of FC data in representing sex differences in a low dimensional representation of functional organization, specifically suggesting that sex differences in the S-A axis may be better represented by FC profiles than FC strength.”

Line 527: Cut “which may underpin cognitive differences” which is overly-speculative.

Line 531: It is not accurate to say that this study “neglected possible effects of gender” since such effects are inherent in the group separation. Just say that you “didn’t test” for moderation by gender-related variables.

Line 540 poor writing: “As such, we capitalized on a binary variable to detect group-level effects, being conscious however that complex social and environmental experiences that go beyond a binary categorization are also likely to have influenced the brains of individuals in our sample throughout their lifetime [61], to an extent that we could however not quantify in this work.”

Could just say that any differences between groups may be attributable to either sex- or gender-based influences (and cite NIH or some other agency that defines these variables).

Line 548: The hippocampal sex difference described in Ref. 6 has not been replicated by most large studies or other meta-analyses. Similarly, the sex differences in amygdala activation reported in Ref. 64 have not been seen other meta-analyses [See Ref. 2, Table 10].

Lines 552-554: “By using the S-A axis, our work identified sex differences embedded in a key macroscale organizational principle that is closely tied to evolutionary expansion and cortical morphometry...” This sentence implies that cortical expansion and morphometry are the basis of the S-A sex difference which is the opposite of what was found, so should be cut.

RESPONSE TO REVIEWERS (NCOMMS-23-59796)

We would like to thank the Editors and Reviewers once again for their positive evaluations, constructive comments, and for the opportunity to submit a second revision of our manuscript. We believe that the resulting changes have further improved the clarity of our work. We have addressed all remaining questions and suggestions in a point-by-point fashion below and we have edited our manuscript and supplementary material accordingly (changes are highlighted in yellow).

Reviewer #1 (Remarks to the Author):

I thank the authors for revising their manuscript. They have now addressed all of my initial concerns and their additional analyses provide further evidence to support their conclusions. I am happy to recommend this manuscript for publication.

Thank you for your positive evaluation.

I do have one additional minor suggestion for the authors. I recommend authors consider using the term assigned sex (or sex assigned at birth) instead of biological sex. I understand prior work with the HCP dataset (and other work in neuroimaging) has used 'biological sex', but recent developments suggest the use of 'assigned sex' and 'sex assigned at birth' is more appropriate (<https://www.ncbi.nlm.nih.gov/pmc/articles/PMC10523819/>).

We have followed your recommendation and now use the term “sex assigned at birth” when referring to the HCP sample.

Reviewer #2 (Remarks to the Author):

I appreciate the authors' thoughtful and thorough response to my previous comments.

I have one last minor comment, based on my initial comment #1 ("The primary finding of the manuscript – that sex differences in S-A are not related to morphometric properties – is informative but also essentially a null result. ..."). I still think including in the manuscript the minimum detectable effect size would be informative, similar to initial GWAS that did as such and shaped our understanding of the expected effect sizes of disease-associated SNPs. However, I acknowledge this may be a field difference and I appreciate that the authors included these effect size calculations in their reviewer response. Particularly given that the reviewer response will be publicly posted online anyway (given Nature Communications' peer review process), I am okay with these calculations not being included in the manuscript.

As a whole, the authors have satisfactorily addressed my comments.

Thank you for your positive evaluation and for your comment on including post hoc computations of minimum detectable effect sizes. After discussing this within the co-

authorship team, we agreed that having this information in the peer review correspondence as you suggested is a good and transparent compromise.

Reviewer #3 (Remarks to the Author):

The authors have appropriately addressed my comments and suggestions.

Thank you for your positive evaluation.

Reviewer #4 (Remarks to the Author):

I am still struggling with the primary aim of this study: although it makes sense to test for a relationship between morphometry and the S-A axis, the reasons for expecting a sex difference in the S-A axis are not well-supported (see below). The last part of the Results, wherein sex differences in functional connectivity profiles are analyzed, is not novel, nor did this analysis actually test whether such FC differences are the basis of sex differences in S-A axis, although it seems obvious they are related given that they are derived using similar functional correlation methods.

Thank you for your additional detailed comments, which have overall helped us further clarify and add nuance to our work, as outlined below.

We are happy to clarify why we hypothesized sex differences in functional organization to be related to differences in cortical morphometry, and why we expected sex differences in the S-A axis, using it as our measure of functional organization.

Our hypothesis of a relationship between sex differences in functional organization and sex differences in cortical morphometry was motivated by one of the most replicated sex differences findings in human neuroscience being that of sex differences in brain size (Ankney (1992) *Nature*; Eliot et al. (2021) *Neuroscience & Biobehavioral Reviews*; Ritchie et al. (2018) *Cerebral Cortex*; Ruigrok et al. (2014) *Neuroscience & Biobehavioral Reviews*; Williams et al. (2021) *Human Brain Mapping*). Various studies have also reported sex differences in functional connectivity –albeit less consistent– primarily in sensory and association areas, i.e., higher functional connectivity in males in sensorimotor regions and in females in the DMN (Ritchie et al. (2018) *Cerebral Cortex*; Shanmugan et al. (2022) *PNAS*; Weis et al. (2020) *Cerebral Cortex*; Allen et al. (2011) *Frontiers in Systems Neuroscience*; Biswal et al. (2010) *PNAS*; Bluhm et al. (2008) *Neuroreport*; Zhang et al. (2020) *Frontiers in Human Neuroscience*; Scheinost et al. (2015) *Human Brain Mapping*). However, no study has tested whether these sex differences in function may be systematically organized and result from sex differences in brain size, specifically sex differences in cortical morphometry. As such, we tested this hypothesis, suggesting that the spatial variability of functional sex differences may reflect changes along a systematic axis of functional organization related to brain size and morphometry, rather than reflecting differences in isolated regions.

Our use of the S-A axis as our measure of functional connectivity was therefore directly motivated by our hypothesis, given that the S-A axis represents a key principle of functional organization that reflects multimodal mechanisms bridging morphometric, structural, and

functional features of cortical organization. In fact, the S-A axis reflects both developmental (Sydnor et al. (2021) *Neuron*) and evolutionary (Valk et al. (2022) *Nature Communications*; Xu et al. (2020) *Neuroimage*) patterns of cortical expansion, aligning with microstructural variation (Burt et al. (2018) *Nature Neuroscience*; Paquola & Hong (2023) *Biological Psychiatry*; Saberi et al. (2023) *PLoS Biology*; Valk et al. (2022) *Nature Communications*). Given that sex differences in functional connectivity have been previously reported, we expected to observe some degree of sex differences in the S-A axis, which is a low dimensional representation of functional connectivity, explaining the most variance and derived from functional connectivity data itself. Furthermore, given that sex differences in functional connectivity have primarily been reported in both sensory and association regions and given that the S-A axis hierarchically differentiates intrinsic functional networks (from sensory to association networks), the S-A axis was also an ideal measure to test our hypothesis of sex differences in the systematic organization of functional connectivity by capturing these distributed differences at the system level.

We have clarified this further in our introduction as follows:

We began by computing the S-A axis as our measure of functional organization, given that it reflects multimodal mechanisms bridging morphometric, structural, and functional features of cortical organization, and given that sex differences in functional connectivity are typically found in regions situated at the extremities of this axis, i.e., in sensory and association regions.

As expected, we found statistically significant sex differences in the S-A axis (23.3% of cortical parcels), with effect sizes ranging between $\beta = 0.210$ (representing a small effect size) and $\beta = 0.435$ (representing a medium effect size). Although we consider this to be a novel and interesting finding, it was only a subcomponent of the broader aim of our study, which was of investigating whether these sex differences in S-A axis loadings were associated with sex differences in cortical morphometry, and which we tested in Figure 3.

Given your comment, we have more clearly reported the magnitude of sex effects on the S-A axis in the main body of our Results section (previously only visible in Figure 1):

We identified sex differences in the S-A axis loadings in 23.3% of cortical parcels (93 out of 400 Schaefer parcels) with small to medium effects (minimum effect size of $\beta = 0.210$; maximum effect size of $\beta = 0.435$), which were distributed across the seven intrinsic functional Yeo networks [39] (Fig. 1B and 1C).

To further nuance our findings and underline that cortex-wide spatial patterns of functional organization are overall very similar between sexes, we have also reported the correlation between the mean functional S-A axes of males and females ($r_{\text{spin}} = 0.996$, $p_{\text{spin}} < .001$) in our results as follows:

Mean cortex-wide spatial patterns of S-A axis loadings were overall highly correlated between the sexes, $r_{\text{spin}} = 0.996$, $p_{\text{spin}} < .001$.

Please note the conceptual distinction between sex differences in mean cortex-wide spatial patterns of S-A axis loadings and regional sex differences in S-A axis loadings. The former – mean cortex-wide spatial patterns in the S-A axis– were expected to be similar between sexes, given that these are group averages and given that the S-A axis as a whole represents a macroscale principle of functional organization, which has been replicated across samples and species such as macaques (Margulies et al. (2016) *PNAS*; Valk et al. (2022) *Nature Communications*) and marmosets (Tong et al. (2022) *Nature Communications*), and to some

lesser extent mice (Huntenburg et al. (2021) *Neuroimage*). The latter –regional sex differences in S-A axis loadings– were expected to mirror the minor, yet statistically significant, sex differences in functional connectivity previously reported in the literature (and resulting as small to medium effect sizes in our study), given that these regional sex differences were tested on S-A axes that were computed at the individual level (rather than group averages).

The last part of the Results exploring the intrinsic functional underpinnings of sex differences in the S-A axis (reported in Figure 4) follows from our findings of no apparent systematic relationship between sex differences in the S-A axis and sex differences in cortical morphometry. We agree that this step appears intuitive but –in the context of our study– is valuable in revealing that different features of the functional connectivity data appear to be differently picked up as sex differences in the S-A axis, which is itself a novel finding. Indeed, sex differences in functional connectivity strength were unexpectedly *not* associated with sex differences in the S-A axis, despite being derived from the same data – as you correctly point out in your comment. On the other hand, the observed sex differences in functional connectivity profiles (despite being minor) coherently reflected the statistically significant sex differences that we observed in network topology (i.e., greater network integration within the DMN in females). Specifically, sex differences in connectivity profiles showed that the strongest functional connections in females seemed to involve the DMN more than in males, whereas males displayed more top connections involving the somatomotor networks relative to females. We believe this provides a valuable and more descriptive picture of the sex differences observed in the S-A axis loadings at the system level and is further consistent with previous findings of greater intrinsic functional connectivity in the DMN in females and in sensorimotor regions in males, whilst still being novel in capturing sex differences from a different angle.

We have edited a passage of our discussion to make this clearer:

We observed greater male dispersion relative to females within the DMN. This finding suggests that areas belonging to the DMN are represented further apart on the S-A axis (i.e., showing less similarity in their FC profiles) in males relative to females, which is also consistent with previous findings of generally more segregated male networks [53]. These network-specific topological sex differences may be related to greater female odds of connections within the DMN, and greater male odds of somatomotor connections with other networks. As such, the apparent sex differences in network topology and functional connectivity profiles –albeit small– provide a more interpretable system-level description of the sex differences observed in the S-A axis loadings, representing a key principle of macroscale cortical organization.

This revision did not answer my overall concern about the significance of sex difference in the S-A axis, which is such a broad way of looking at the cerebral cortex that it is challenging to relate to any of the well-studied behavioral and clinical group-level differences between women and men. By definition, the S-A axis includes all of the cerebral cortex except for motor areas. Although expansion of this axis may be important in evolution and development, the authors do not make a convincing case for why they expect it to differ between men and woman. And while they spend much of the Introduction (lines 68-99) arguing that sex differences in regional cortical volumes are consistent with sex difference in the overall S-A axis, these regional differences have proven markedly discrepant across recent studies (see Line 93 comment below). So this is not a strong rationale for their experiment, and its actual result finding (Fig. 3) is consequently less interesting.

We hope that our answer to your first point above could further clarify our motivation to study associations between sex differences in the S-A axis and sex differences in cortical morphometry, as well as why we expect sex differences in S-A axis loadings.

To clarify the few additional points that you touch on:

The S-A axis by definition includes all of the cerebral cortex, including motor areas, as it is computed by reducing the dimensionality of functional connectivity data in all of the regions of the cortex.

We appreciate your point that the S-A axis may at first appear to be a broad and abstract measure of cortical organization. However, in addition to the abovementioned multiscale and multimodal associations, S-A axis has been receiving growing support as a meaningful measure capturing individual differences and with specific associations to behavioral and clinical measures – e.g., semantic cognition (Shao et al. (2022) *Cortex*), autism diagnosis (Hong et al. (2019) *Nature Communications*), aging and measures of general cognition (i.e., fluid and crystallized intelligence; Knodt et al. (2023) *Human Brain Mapping*). We would like to underline that these behavioral and clinical associations are not what we are trying or claiming to establish in our study, but add credibility to the general use of the S-A axis as a meaningful measure of functional organization. Specifically important to our rationale is the fact that the S-A axis can conceptually be understood as a measure to probe the cortical hierarchy, differentiating sensory from association regions. Such a differentiation is anchored in functional connectivity, microstructure, and patterns of brain expansion, as mentioned earlier. We believe that it is not sufficient to interpret cortical regions and networks in isolation. Instead, they should be viewed within a global organizational framework, as done in our study. Specifically, we examine how sex differences in functional connectivity manifest in broader patterns of functional organization, using the S-A axis. In doing so, our investigation aims to integrate local observations of sex differences in intrinsic function into a system-level framework. This framework is what we then use to examine associations between sex differences in function and sex differences in cortical morphometry.

Another concern is that this study seems to over-hype what is actually a very modest sex difference. On Line 349 it is noted that only 1.25% of the 160,000 functional connections showed a significant sex difference. So it seems a stretch to call this a “divergent system-level organization of functional networks” between males and females [Line 573]. It’s also a far extrapolation to suggest that the findings in this paper will contribute meaningfully to understanding sex and gender differences in psychiatric vulnerability [lines 581-583].

Indeed, sex effects in functional connections were modest – we have revised a sentence of our discussion accordingly:

Instead, we found that sex differences in the S-A axis were related to modest sex differences in FC profiles, which also presented qualitative sex differences in the proportional breakdown of networks involved.

Please note that when referring to the “divergent system-level organization of functional networks”, we refer to the statistically significant sex differences in our measure of network topology, specifically greater male within-network dispersion in the DMN. We have revised this sentence of the discussion to make our claim more specific to the observed effect:

Our findings instead suggest that sex differences in the S-A axis are, to some extent, intrinsically related to differences in FC profiles and network topology. Although these sex differences appear to be small, they may be meaningful for broader sex differences in functional cortical organization, and future research should explore factors driving males and females to form these few distinct functional connections that are associated with sex differences in the system-level organization of functional networks, notably of the DMN.

As supported by others in the field of sex differences, “studying how biological sex contributes to our health can help understanding of disease etiology, manifestation, progression, and treatment of disease” (Galea et al. (2020) *Frontiers in Neuroendocrinology*), and we believe that an important contribution to this begins with studying sex as a biological variable in fundamental neurobiological research on healthy individuals (see also Shansky & Murphy (2021) *Nature Neuroscience*). However, we revised the last sentences of our discussion given the confusion that they caused with regard to the direct implications of our study findings (originally mentioning the importance of studying sex differences in the brain for psychiatry) as follows:

Generally, it is crucial for research in neuroscience to systematically test for sex differences in brain structure and function, as well as their biological and environmental underpinnings, in order to produce more rigorous and representative findings, ultimately leading to a more translational body of knowledge [70].

As requested in my prior review, it’s important to state the many dimensions in which environmental factors might drive male and female functional networks to differ, beyond just “stress” [line 577]. I recommended then and reiterate my request that the authors report (in the Supplement) the male and female averages for any of the following factors which are moderated by gender and could be influencing the results: 1) income, 2) years of education, 3) physical activity level, and 4) caregiver/parental status, or any other variables that HCP collects which may differentially affect brain health or function in 29 year-old men vs. women.

We apologize for having only partially addressed your original request of reporting demographic variables in our sample, which we did in our previous revision without however specifying the breakdown by sex. We have now edited the Supplementary Table 3 (which includes the sociodemographic data that is available in the HCP sample) to include a breakdown by sex as follows:

Race							
White	Black/AM	Asian/NH/OPI	AI/AN	N/A	>1		
752 (392)	142 (87)	63 (33)	2 (1)	17 (10)	24 (13)		
Ethnicity							
Not Hispanic/Latino		Hispanic/Latino			N/A		
896 (491)		91 (39)			13 (6)		
Employment							
Full-time		Part-time			Not working		
672 (326)		177 (110)			149 (99)		
Total household income							
1	2	3	4	5	6	7	8
69 (43)	72 (32)	132 (60)	118 (67)	99 (56)	207 (115)	139 (78)	157 (81)
Education (years)							
<11	12	13	14	15	16	17<	
32 (18)	134 (73)	62 (28)	123 (64)	61 (20)	434 (237)	152 (95)	

Table 3. Sociodemographic breakdown of Human Connectome Project sample. Number of subjects within each category (number of females in parentheses). AM, African American; NH, Native Hawaiian; OPI, Other Pacific Island; AI, American Indian; AN, Alaskan Native; N/A, unknown or not reported; >1, more than one race. Total household income key: 1 = <\$10k, 2 = 10k-19'999, 3 = 20k-29'999, 4 = 30k-39'999, 5 = 40k-49'999, 6 = 50k-74'999, 7 = 75k-99'999, 8 = >100k.

We agree that there are many environmental dimensions that, on average, may differ between sexes/genders. Yet, the purpose of those few lines in our discussion is not to list relevant sex and gender differences in environmental factors or cognition (as we do not believe that we would adequately be able to capture them in a few lines), but rather to underline the general importance –at a more conceptual level– of considering that there may not only be sex differences in environmental exposures but also sex differences in the mechanisms by which the “same” environmental factors may affect brain function. The latter is greatly understudied and exemplifies the complexity of what “moderation by sex” may look like. For this, we therefore use stress as an example given that the physiological stress response and related sex differences have been widely researched, with studies specifically reporting sex differences in the effects of the stress response on brain function – further revealing mechanistic pathways through which this moderation by sex occurs, i.e., epigenetic mechanisms. To further clarify this, we have revised the passage about environmental factors in the discussion as follows:

Environmental factors should equally be considered, as they may not only differ on average as a function of sex and gender, but may also differently affect brain function across the sexes through divergent mechanisms [69]. An example of this is stress, whereby sex differences in the stress response have been found to contribute to sex differences in brain function and psychopathology via epigenetic mechanisms [69-71].

Finally, I found the writing to be somewhat convoluted, more so than the prior draft, leading to a lack of clarity in the aims and significance of the findings. I've indicated a few examples below to illustrate this.

Thank you for these specific suggestions to improve the clarity of our work – we have revised our manuscript accordingly.

Specific comments:

Line 23: It's hard to see brain size as “robustly” different between the sexes, given that the distributions of TBV overlap by nearly 50% between females and males (per Ref 5).

Thank you for your comment. We use the term robust referring to the fact that sex differences in brain size are statistically significant and widely replicated. The effect size of this difference actually appears to be large – citing the commentary by Eliot (2024; *Biology of Sex Differences*) that you refer to and share in your next comment: “While a statistically large effect ($d = 1.31$ in the largest study to-date; [2]), brain volume nonetheless overlaps by 51% between female and male distributions, so not a “categorical” difference or “dimorphism”—that is, like the difference between ovaries and testes, or the tail of a peacock versus peahen”. Here, they underline the fact that sex differences in brain size do not make the brain a dimorphic organ, but they still acknowledge the importance of this difference in terms of effect size and statistical significance. However, to address your concerns and avoid any confusion about claiming dimorphism of the brain, we have changed the first sentence of our abstract (originally “Brain size robustly differs between sexes.”) as follows:

Differences in brain size between the sexes are consistently reported.

Line 93: According to a more recent paper by Eliot (<https://doi.org/10.1186/s13293-024-00585-4>) the regional brain volume sex differences reported by Williams et al. [Ref. 7] and other studies (including Refs. 6 and 18) have not been well-replicated, including the anterior and posterior cingulate, precuneus, inferior and middle frontal gyri, insula and lateral occipital cortex. But such claims are relied upon to rationalize this study, which states on Line 93 that such “findings further depict the apparent relevance of the S-A axis as an axis of morphometric variability between the sexes.”

Thank you for sharing this recent commentary. We have now further clarified in our manuscript that sex differences in regional brain volumes are generally small and variable:

In fact, although local sex differences in cortical morphometry are typically small in size and vary across studies [18], they have been reported in both sensory and association regions: A meta-analysis identified volumetric sex differences in multiple cortical regions, including the anterior and posterior cingulate gyri, precuneus, right frontal pole, inferior and middle frontal gyri, insular cortex, Heschl's gyrus, and lateral occipital cortex [6].

Please also see our responses to your previous comments with respect to the broader rationale of our study in using the S-A axis. We hope this clarifies how we interpret the link between these variable local findings and our system-level approach.

Line 449: Example of convoluted writing: “Indeed, given that sensory and association regions are respectively known to make short- and long-range connections [25], it is expectable to observe sex differences in mean geodesic distances and some level of a sex by mean geodesic distance interaction effects on the S-A axis loadings given the observed sex differences in FC profiles involving these specific regions. As such, our findings highlight the variable relevance of different features of FC data in representing sex differences in a low dimensional representation of functional organization, specifically suggesting that sex differences in the S-A axis may be better represented by FC profiles than FC strength.”

We agree that the writing is convoluted here, apologies for this. We have revised this passage as follows:

This is also consistent with the few regional statistically significant interaction effects of sex by mean geodesic distance of FC profiles on S-A axis loadings. Indeed, given that sensory and association regions are respectively known to have primarily short- versus a mix of short- and long-range connections [26], sex differences in FC profiles involving regions in the somatomotor network and the DMN may be related to differences in the geodesic distances of functional connectivity profiles between sexes. As such, our findings suggest that sex differences in the S-A axis may be better represented by sex differences in FC profiles, i.e., the topology of functional connections, than sex differences in FC strength alone.

Line 527: Cut “which may underpin cognitive differences” which is overly-speculative.

We have removed this statement as requested.

Line 531: It is not accurate to say that this study “neglected possible effects of gender” since such effects are inherent in the group separation. Just say that you “didn’t test” for moderation by gender-related variables.

We agree and have revised the sentence as follows:

Firstly, by focusing on sex assigned at birth, we did not test for effects of gender or gender-related variables on functional organization and its morphometric correlates. This is relevant given that there are multiple factors contributing to sex/gender differences, including both biological and social factors.

Line 540 poor writing: “As such, we capitalized on a binary variable to detect group-level effects, being conscious however that complex social and environmental experiences that go beyond a binary categorization are also likely to have influenced the brains of individuals in our sample throughout their lifetime [61], to an extent that we could however not quantify in this work.” Could just say that any differences between groups may be attributable to either sex- or gender-based influences (and cite NIH or some other agency that defines these variables).

We have revised this sentence, making some additional changes to the sentence preceding it, as follows:

Nevertheless, we intentionally focused on the dichotomous variable of sex to identify group-level effects, as our study aimed to investigate the correspondence (i.e., shared variance) between sex differences in cortical morphometry and sex differences in cortical functional organization, regardless of variance

explained by gender. We recognize the limitations of using a binary variable given that differences between groups may be attributable to both sex- and gender-based influences [62]. However, quantifying these influences was beyond the scope of this study.

Line 548: The hippocampal sex difference described in Ref. 6 has not been replicated by most large studies or other meta-analyses. Similarly, the sex differences in amygdala activation reported in Ref. 64 have not been seen other meta-analyses [See Ref. 2, Table 10].

The references that you are referring to pertain to our second limitation, highlighting that our work focused on neocortical functional organization, excluding subcortical structures, where sex differences in structure and function have been previously reported. We believe that the articles that we have cited (both being meta-analyses) are sufficient for this purpose, but have clarified in text that these effects show some variability:

For example, the amygdala and hippocampus have shown a variable degree of both structural [6] and functional [65] sex differences.

Lines 552-554: “By using the S-A axis, our work identified sex differences embedded in a key macroscale organizational principle that is closely tied to evolutionary expansion and cortical morphometry...” This sentence implies that cortical expansion and morphometry are the basis of the S-A sex difference which is the opposite of what was found, so should be cut.

The sentence that you are referring to serves the purpose of underlining the novelty of our study in using the S-A axis. We have revised this sentence to avoid misunderstandings as follows:

By using the S-A axis, our work took a system-level approach to identify sex differences embedded in a key macroscale organizational principle, going beyond previous research on functional differences between the sexes solely focusing on intrinsic brain function.